# Learning What to Generate: A Reinforcement Learning-based Closed-Loop Augmentation Framework for Person Re-identification

**Xincheng Shi** [1]  **Changxiao Ma** [1]  **Yongfei Zhang** [1,2]  **Yuzhuo Ma** [1]  **Rongye Shi** [1]

## Abstract

Person re-identification (ReID) models are sensitive to long-tail nuisances (e.g., rare viewpoints, occlusions, complex backgrounds), yet current generative augmentation is largely *open-loop*: prompts/conditions are sampled heuristically without verifying whether the synthesized samples improve ReID discriminability. We introduce **ReasonAug**, a **closed-loop** framework that learns an *image-conditioned instruction policy* for a **frozen** generator, turning augmentation into a sequential decision problem over instruction tokens. A Semantic Reasoning Agent (SRA) performs **hierarchical planning** from global semantics to identity-critical local cues, producing structured edit instructions whose utility is *verified* by downstream ReID feedback. To make closed-loop optimization reliable, we propose **Metric-Aligned Gated Reward (MAGR)**, which converts metric-learning objectives into a dense reward while *gating* task shaping by identity preservation to prevent reward hacking, and **Structure-Aware Entropy (SAE)**, which allocates exploration *per token* to lock identity-critical cues while diversifying nuisance factors. Experiments on Market-1501 and MSMT17 demonstrate state-of-the-art performance, confirming that closing the augmentation loop and learning what to generate yield more discriminative training data than open-loop alternatives.

## 1. Introduction

Deep learning-based Person Re-identification (ReID) is fundamentally data-hungry (Ye et al., 2021; Zheng et al., 2015;

[1]School of Computer Science and Engineering, Beihang University, Beijing, China [2]State Key Laboratory of Virtual Reality Technology and Systems, Beihang University, Beijing, China. Correspondence to: Yongfei Zhang <yfzhang@buaa.edu.cn>.

*Proceedings of the 43$^{rd}$ International Conference on Machine Learning*, Seoul, South Korea. PMLR 306, 2026. Copyright 2026 by the author(s).

Wei et al., 2018). While recent diffusion-based generation and editing methods—leveraging Diffusion Models (Ho et al., 2020; Dhariwal & Nichol, 2021) with structural controls like ControlNet (Zhang et al., 2023) or reference-based editing (e.g., Animate Anyone) (Hu, 2024)—have achieved impressive photorealism and controllability, they are largely designed to maximize visual fidelity rather than downstream ReID performance. In practice, these generators are used as an *open-loop* augmentation process: prompts and conditions are sampled by static heuristics to create "more" data (Zhong et al., 2018; Zheng et al., 2017; Bhunia et al., 2023; Niu et al., 2025), without feedback on whether the synthesized samples actually improve embedding discriminability or preserve identity-critical cues. As a result, blindly commanding a powerful generator is inefficient. We identify two key limitations in current pipelines: (1) **The Instruction Gap**: Tools like ControlNet are excellent *executors* but poor *planners* (Zhang et al., 2023). They can render complex scenes but cannot reason *whether* a specific variation contributes to the ReID decision boundary. (2) **The Imitation Trap**: Heuristic prompting strategies merely mimic the source distribution, failing to synthesize *hard negatives* or *corner cases* that are rare in the training data. As illustrated in Fig. 1, this "policy gap" manifests even with strong instruction-following editors: without task-aware reasoning, generated samples can drift in identity-critical details and waste generation budget on uninformative variations. This motivates a shift from "how to generate" (visual fidelity) to "what to generate" (semantic planning). We propose **ReasonAug**, a **closed-loop** data augmentation framework that learns an *instruction policy* for a *frozen* editor with verifiable feedback from the downstream ReID objective. Concretely, a Semantic Reasoning Agent (SRA) observes a reference image and proposes edit instructions; the frozen generator executes them; a ReID model evaluates the generated sample and provides rewards; and the policy is optimized via RL with verifiable rewards (RLVR), closing the loop between generation and training utility.

To make the instruction policy *task-effective* rather than merely descriptive, we introduce a **CoT-style hierarchical planning** procedure that explicitly reasons from *global semantics* to *identity-critical local cues* (e.g., unique textures, logos, accessories) before compiling an executable

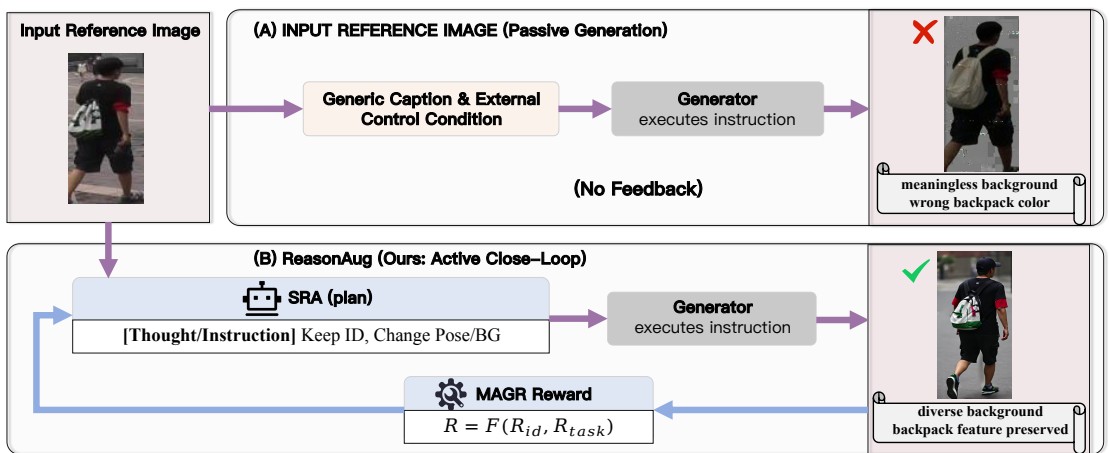

*Figure 1.* **Comparing open-loop vs. closed-loop augmentation paradigms for ReID.** (a) Passive open-loop methods rely on generic prompts without feedback, often leading to identity drift or low-utility samples (e.g., wrong backpack color). (b) **ReasonAug** introduces an active closed-loop framework where a Reasoning Agent (SRA) learns to plan optimal edits guided by task-specific rewards (MAGR), ensuring both high semantic diversity and identity preservation.

edit instruction. This structured reasoning helps the policy preserve identity while allocating variation to nuisance factors such as pose, background, and illumination.

Finally, closed-loop optimization is unstable without careful exploration control and reward shaping. We address this with **Metric-Aligned Gated Reward (MAGR)** for dense, identity-safe metric-learning rewards that prevent reward hacking, and **Structure-Aware Entropy (SAE)** for token-level stability–diversity separation.

**Our contributions are as follows:**

- **Closed-loop augmentation framework.** We formulate generative ReID augmentation as **closed-loop instruction policy learning**: the generator remains frozen, while the policy is optimized with verifiable ReID feedback to learn *what to generate*.

- **CoT-style hierarchical planning.** We propose a structured reasoning procedure that plans edits from *global semantics* to *identity-critical local cues*, producing targeted instructions that preserve identity while increasing semantic diversity.

- **Metric-Aligned Gated Reward (MAGR).** We design a dense, metric-learning-aligned reward with identity gating to prevent reward hacking and reliably drive discriminability improvements.

- **Structure-Aware Entropy (SAE).** We introduce token-level entropy allocation that *locks* identity-critical tokens and *encourages* exploration on variation tokens, stabilizing closed-loop RL.

Experiments on Market-1501 and MSMT17 show that ReasonAug achieves **state-of-the-art** performance (89.0% /

90.6% mAP on Market-1501 with TransReID / CLIP-ReID).

## 2. Related Work

### 2.1. Generative Data Augmentation for ReID

Data augmentation for ReID ranges from standard perturbations (e.g., random erasing (Zhong et al., 2020)) to generative pipelines that reduce camera/domain gaps via GANs and unpaired translation (e.g., SPGAN (Deng et al., 2018) and CycleGAN-based methods (Zhu et al., 2017; Liu et al., 2024b)) and, more recently, diffusion-based synthesis with improved fidelity and diversity (Tao et al., 2024; Dai et al., 2025). Despite this progress, most approaches remain *open-loop*: they generate samples using static heuristics without learning which semantic variations most improve downstream ReID. ReasonAug instead treats augmentation as **instruction policy learning**—learning *what to generate* under verifiable ReID feedback.

### 2.2. Controllable Generation and Editing

Structure-aware generation introduces explicit conditions such as pose (e.g., Pose-dIVE (Kim et al., 2024)) or 3D priors to improve controllability. Meanwhile, diffusion-based editors like InstructPix2Pix (Brooks et al., 2023) and HIVE (Zhang et al., 2024) achieve strong instruction-following capabilities. However, these methods either rely on predefined augmentation strategies or focus solely on visual fidelity. They lack a mechanism to decide *what* to generate for maximizing downstream discriminability. In contrast, we keep the editor frozen and learn a ReID-specific instruction policy that plans *task-effective* edits via **CoT-style hierarchical planning**, stabilized by **Structure-Aware Entropy (SAE)**.

### 2.3. Vision-Language Models and RL Reasoning Agents

Vision-language models (VLMs) have evolved from representation learning to general-purpose multimodal reasoning systems. CLIP (Radford et al., 2021) provides strong contrastive image-text features, and instruction-tuned LVLMs such as LLaVA (Liu et al., 2023) enable conversational multimodal understanding. Recent visual/multimodal CoT studies show that explicitly externalizing intermediate steps—often grounded in visual regions or sketches—can improve performance and robustness on challenging multimodal reasoning and grounding tasks (Hu et al., 2024; Xu et al., 2025a; Gao et al., 2025; Man et al., 2025). The Qwen-VL family further improves fine-grained perception and robustness: Qwen2-VL introduces dynamic-resolution visual tokenization (Wang et al., 2024a), and Qwen2.5-VL strengthens localization and long-context multimodal reasoning (Bai et al., 2025b; Xu et al., 2025b). Meanwhile, RLHF/RLVR has become a standard alignment recipe, from PPO-style optimization (Schulman et al., 2017) to group-based methods such as GRPO (Shao et al., 2024), with recent scalable variants including DAPO (Yu et al., 2025) and GSPO (Zheng et al., 2025); multimodal RL reasoning is also emerging (e.g., ReasonGen-R1 (Zhang et al., 2025), BLIP3o-Next (Chen et al., 2025)) and AR-GRPO (Yuan et al., 2025b). Our work connects these trends by optimizing a strong VLM into a decision-making agent for ReID augmentation, using **Metric-Aligned Gated Reward (MAGR)** and SAE for stable *analysis-by-synthesis*.

## 3. Method

### 3.1. Framework

Fig. 2 overviews ReasonAug. Concretely, ReasonAug consists of three components: (i) a Semantic Reasoning Agent (SRA; Fig. 2(a)) that produces CoT-style plans and an executable edit instruction; (ii) a Text Alignment Refiner (TAR) that adapts the instruction to the frozen editor; and (iii) GRPO-based optimization under Metric-Aligned Gated Reward (MAGR; Fig. 2(b)) and Structure-Aware Entropy (SAE; Fig. 2(c)) (Secs. 3.1.1–3.4). Our goal is to learn an instruction policy that guides a *frozen* image generator to synthesize training samples that improve a downstream ReID model. Given a reference image $I_{\text{ref}}$ of identity $y$, the policy $\pi_\theta$ autoregressively generates an instruction sequence $y_{1:L} \sim \pi_\theta(\cdot \mid I_{\text{ref}})$ (including optional chain-of-thought and a final executable instruction). We denote the executable instruction segment as $y^{\text{inst}}$. A frozen generator $G$ (Sec. 3.1.1) then produces an edited image

$$I_{\text{gen}} = G(I_{\text{ref}}, \text{TAR}(y^{\text{inst}})), \qquad (1)$$

where TAR is a lightweight text adapter. We evaluate $I_{\text{gen}}$ with a frozen ReID encoder and compute a scalar reward $R$ (Sec. 3.3). The learning problem is thus a decision process

where actions are instruction tokens, the environment is the frozen generator plus reward computation, and the objective is to maximize expected reward:

$$\max_\theta \ \mathbb{E}_{y_{1:L} \sim \pi_\theta(\cdot|I_{\text{ref}})} \left[ R(I_{\text{ref}}, I_{\text{gen}}, y) \right]. \qquad (2)$$

Crucially, we do *not* train the generator; we optimize only the instruction policy (and TAR), making the system modular and robust to future generator upgrades.

#### 3.1.1. FROZEN GENERATOR AND TAR

We keep a pretrained instruction-following generator $G$ **frozen** to preserve its priors. Since ReID requires attribute-dense prompts that can be out-of-distribution for generic editors, we use a lightweight **Text Alignment Refiner (TAR)** to map the executable instruction $y^{\text{inst}}$ into the generator's preferred conditioning space. TAR is small ($\sim$2.3M parameters) and is pre-trained with paired ReID data while keeping $G$ fixed.

#### 3.1.2. POLICY OPTIMIZATION WITH GRPO

We optimize the SRA using Group Relative Policy Optimization (GRPO) (Shao et al., 2024). We do not claim a new RL optimizer; GRPO is chosen as a practical optimizer for our VLM-based policy because it avoids the extra critic/value network required by PPO and reduces the variance of vanilla policy-gradient updates through group-relative reward normalization. This keeps the trainable footprint limited to the SRA/TAR adapter while preserving stable learning in the large action space of text instructions. For each $I_{\text{ref}}$, we sample a group of $G$ instruction candidates $\{y^{(i)}\}_{i=1}^G$ from the current policy and obtain rewards $\{r_i\}$. GRPO computes a group-relative advantage:

$$\hat{A}_i = \frac{r_i - \mu_G}{\sigma_G + \epsilon}, \qquad (3)$$

where $\mu_G = \frac{1}{G} \sum_{j=1}^G r_j$, $\sigma_G = \sqrt{\frac{1}{G} \sum_{j=1}^G (r_j - \mu_G)^2}$

The clipped GRPO objective follows PPO-style updates:

$$\mathcal{L}(\theta) = -\frac{1}{G} \sum_{i=1}^G \sum_{t=1}^{L_i} \min\left( \rho_{t,i} \hat{A}_i, \ \text{clip}(\rho_{t,i}, 1-\epsilon, 1+\epsilon)\hat{A}_i \right), \qquad (4)$$

where $\rho_{t,i}$ is the importance ratio.

### 3.2. CoT-style Hierarchical Planning

The Semantic Reasoning Agent (SRA) is a multimodal LLM (e.g., Qwen2-VL) that maps $I_{\text{ref}}$ to an instruction. Rather than plain captioning, we use **CoT-style hierarchical planning** to explicitly reason from *global semantics* (scene, pose, camera) to *identity-critical local cues* (e.g., unique textures, logos, accessories), before producing the executable instruction $y^{\text{inst}}$. Concretely, the planning trace decomposes into:

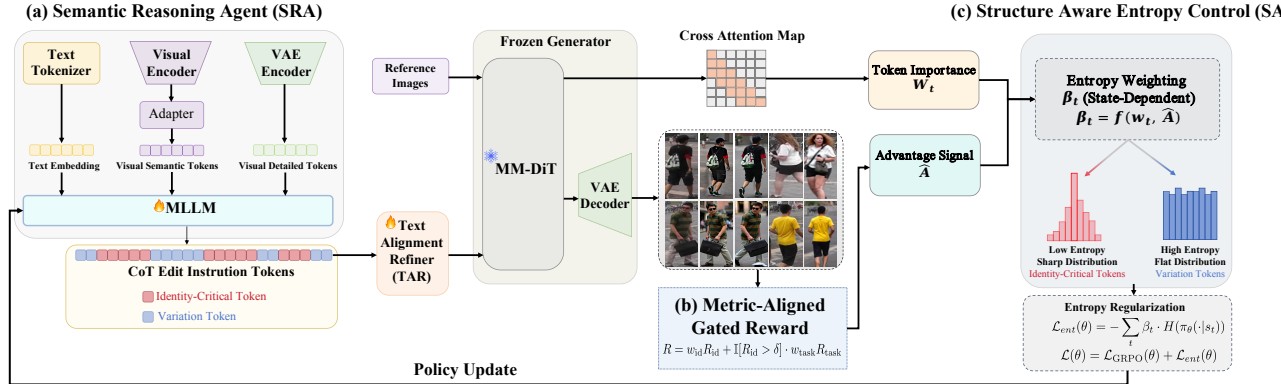

*Figure 2.* **Overview of ReasonAug.** The generator remains **frozen**; we learn only the **instruction policy**. The SRA uses **CoT-style hierarchical planning** (global → local cues) to produce edit instructions, refined by TAR. The policy is optimized via GRPO with **MAGR** (identity-gated rewards) and **SAE** (token-level entropy control), closing the loop between generation and ReID feedback.

(i) global description, (ii) identity-critical cue list, (iii) allowed variation set, and (iv) a concise edit instruction. This structure improves identity preservation while allocating diversity to nuisance factors.

**SFT warm-start.** We warm-start SRA via supervised fine-tuning on curated tuples $\{I_{\text{ref}}, \text{CoT+instruction}, I_{\text{target}}\}$ to learn the format and basic attribute recognition. During SFT, the model is trained for next-token prediction from $I_{\text{ref}}$ to the CoT/instruction sequence; $I_{\text{target}}$ is used only for curating supervision and is not provided as an input to the policy. We then apply RL to align the policy with downstream ReID rewards, enabling the agent to plan *task-effective* variations beyond imitation.

### 3.3. Metric-Aligned Gated Reward (MAGR)

RL on generative models is prone to (i) **reward hacking** and (ii) **sparse / uninformative rewards**. We address both by translating the triplet objective from metric learning into a dense RL reward with an identity gate. See Fig. 2(b) in the framework and Fig. 10 in Appendix in detail for an illustration.

**Frozen encoder and distances.** Let $f(\cdot)$ be a frozen ReID encoder producing $\ell_2$-normalized embeddings and $d(u, v) = 1 - u^\top v$ be cosine distance. **Crucially, we use ResNet50-IBN (Pan et al., 2018) as the reward encoder, which is *independent* of the downstream evaluation models (TransReID, CLIP-ReID).** This design choice ensures that the learned policy captures general visual semantics rather than overfitting to a specific network's feature space. Given $I_{\text{gen}}$, we define anchor $a = f(I_{\text{gen}})$. We mine the hardest positive $p^*$ from same-ID samples and hardest negative $n^*$ from different IDs using a two-layer memory bank (below).

**Identity preservation reward.** We define the identity re-ward as the cosine similarity between reference and generated embeddings:

$$R_{\text{id}} = f(I_{\text{ref}})^\top f(I_{\text{gen}}) = 1 - d(f(I_{\text{ref}}), f(I_{\text{gen}})). \quad (5)$$

This provides a continuous signal measuring how well the generated image preserves the identity of the reference.

**Metric shaping reward.** We define:

$$R_{\text{task}} = \tanh\left(\frac{d(a, n^*) - d(a, p^*) + m}{\tau}\right), \quad (6)$$

which is monotone in the triplet margin and provides non-zero gradients even when the margin is satisfied.

**Identity gating.** To prevent degenerate artifacts that artificially increase metric distances, we activate $R_{\text{task}}$ only when identity preservation is reliable:

$$R = w_{\text{id}}R_{\text{id}} + \mathbb{I}[R_{\text{id}} > \delta] \cdot w_{\text{task}}R_{\text{task}}. \quad (7)$$

When $R_{\text{id}} \leq \delta$, the policy receives only identity-correction feedback; once identity is correct, the reward switches on discriminability shaping.

**Two-layer memory bank for hard mining.** We maintain (i) a static FAISS prior that stores top-$K$ hard negative identities per ID, and (ii) a dynamic queue of recent generated embeddings per identity. This yields $O(BK)$ online mining and naturally induces a curriculum: as the policy improves, the dynamic bank becomes harder.

### 3.4. Structure-Aware Entropy (SAE)

GRPO with uniform entropy regularization suffers from the *uniform entropy paradox*: a single global coefficient cannot simultaneously *lock* identity-critical cues and *explore* nuisance-factor variations. To resolve this dilemma, we propose **Structure-Aware Entropy (SAE)**, which allocates

entropy *per-token* using intrinsic rollout signals (Fig. 2(c)). Intuitively, SAE acts as a dynamic traffic light: it shows a "Red Light" (low entropy) to identity-critical tokens to freeze them, and a "Green Light" (high entropy) to variation tokens to encourage exploration.

**Token Sensitivity from Cross-Attention.** To decide where entropy should be constrained, we require a proxy for token importance. We leverage the frozen editor's cross-attention maps, which provide effective token-to-pixel attribution in diffusion models. Prior work shows that aggregated cross-attention correlates with semantic grounding and can be used to explain or localize generated content (Tang et al., 2023; Liu et al., 2024a; Wang et al., 2024b). Concretely, we define a token sensitivity score $S_t$ by spatial and layer-wise pooling of cross-attention responses:

$$S_t = \text{Pool}_{\text{spatial,layers}}(\text{AttnMap}_t). \quad (8)$$

We then select *critical* tokens via a quantile threshold $\kappa$:

$$w_t = \mathbb{I}\Big[S_t \geq \text{Quantile}_\kappa(\{S_\tau\}_{\tau=1}^L)\Big]. \quad (9)$$

**Outcome-Conditioned Entropy Allocation.** Besides sensitivity, we use the group-relative advantage $\hat{A}$ from GRPO as a quality signal. SAE assigns a dynamic entropy coefficient $\beta_t$:

$$\beta_t = w_t \cdot \beta_{\text{adaptive}}(\hat{A}) + (1 - w_t) \cdot \beta_{\text{neutral}}, \quad (10)$$

where the adaptive term switches modes according to whether the current rollout preserves identity:

$$\beta_{\text{adaptive}}(\hat{A}) = \begin{cases} \beta_{\text{lock}} \approx 0 & \hat{A} > 0, \\ \beta_{\text{explore}} > 0 & \hat{A} \leq 0. \end{cases} \quad (11)$$

The resulting entropy regularizer is

$$\mathcal{L}_{\text{ent}}(\theta) = -\sum_{t=1}^L \beta_t \cdot H(\pi_\theta(\cdot|s_t)). \quad (12)$$

**Gradient Dynamics.** Let $z_k$ denote the logit for token $k$ and $p_k$ its probability. The entropy gradient satisfies $\nabla_{z_k}\mathcal{L}_{\text{ent}} \propto -\beta_t\, p_k(-\log p_k - H)$, yielding two behaviors: (i) when $\beta_t \approx 0$ (**Locking**), the entropy term vanishes and the policy can sharpen around successful identity-critical tokens; (ii) when $\beta_t > 0$ (**Exploration**), the policy is penalized for low entropy, flattening the distribution to encourage alternative edits.

**Validity.** Per-token entropy is well-defined since autoregressive sequence entropy decomposes as $H(\mathbf{y}) = \sum_t H(y_t|y_{<t})$. Moreover, weighting by $w_t$ mitigates credit assignment errors: tokens with high cross-attention sensitivity are more causally responsible for image-level outcomes, making them the appropriate locations to modulate exploration.

---

**Algorithm 1** Training ReasonAug (SFT $\rightarrow$ RL with MAGR + SAE)

---

1: **Input:** Reference images $\{I_{\text{ref}}^i\}$, SFT-initialized policy $\pi_\theta$
2: **// Stage 1: SFT Warm-start**
3: **for** each SFT sample **do**
4:     Update $\theta$ via next-token prediction loss
5: **end for**
6: **// Stage 2: RL Alignment**
7: **for** iteration $k = 1$ to $K$ **do**
8:     Sample group $\{y^{(i)}\}_{i=1}^G \sim \pi_\theta(\cdot \mid I_{\text{ref}})$
9:     Generate $\{I_{\text{gen}}^{(i)}\}$ via frozen generator $G$ (with TAR); extract cross-attention maps
10:     Compute MAGR rewards $\{r_i\}$ (Sec. 3.3) and GRPO advantages $\{\hat{A}_i\}$ (Sec. 3.1.2)
11:     Compute SAE coefficients $\{\beta_t\}$ from attention and advantage (Sec. 3.4)
12:     Update $\theta$ by minimizing Eq. (13)
13: **end for**

---

### 3.4.1. TOTAL OBJECTIVE AND TRAINING

We combine the GRPO loss, SAE, and a KL penalty to the SFT reference policy:

$$\mathcal{L}_{\text{Total}}(\theta) = \mathcal{L}_{\text{GRPO}}(\theta) + \mathcal{L}_{\text{ent}}(\theta) + \gamma D_{\text{KL}}(\pi_\theta \| \pi_{\text{ref}}). \quad (13)$$

Algorithm 1 summarizes the full pipeline.

## 4. Experiments

### 4.1. Experimental Setup

**Datasets.** We evaluate on Market-1501 (Zheng et al., 2015) and MSMT17 (Wei et al., 2018). Unless otherwise specified, we generate 4 augmented images per identity.

**Metrics.**

- **Downstream ReID Performance:** We report mAP and Rank-1 accuracy using TransReID (He et al., 2021) and CLIP-ReID (Li et al., 2023) as backbones.

- **Generation Quality & Diversity:** We report **FID** for image fidelity and **Identity Score** (cosine similarity) for ID preservation. To quantify semantic diversity, we report **Intra-ID LPIPS** (denoted as **Div.**). A higher Div. score indicates the policy successfully explores diverse poses and backgrounds rather than collapsing to similar modes.

- **Sample Efficiency:** The downstream performance gain per generated image (see Table 8).

*Table 1.* Quantitative performance comparison for pedestrian generation using the **Market-1501** dataset. Best results are in **bold**.

| Training Data | TransReID | | CLIP-ReID | |
|---|---|---|---|---|
| | mAP ↑ | R-1 ↑ | mAP ↑ | R-1 ↑ |
| Market-1501 (Base) | 87.3 | 94.3 | 89.8 | 95.5 |
| DG-Net (Zheng et al., 2019) | 75.8 | 89.3 | 86.6 | 94.4 |
| DPTN (Zhang et al., 2022) | 82.6 | 92.2 | 85.3 | 93.3 |
| Pose2ID (Yuan et al., 2025a) | 87.4 | 94.5 | 89.4 | 95.0 |
| OmniPerson (Ma et al., 2025) | 88.7 | 94.7 | 90.4 | 95.5 |
| **ReasonAug (Ours)** | **89.0** | **94.9** | **90.6** | **95.5** |

*Table 2.* Data augmentation performance on **MSMT17**. The complex variations in MSMT17 highlight the advantage of our reasoning-guided diversity.

| Training Data | TransReID | | CLIP-ReID | |
|---|---|---|---|---|
| | mAP ↑ | R-1 ↑ | mAP ↑ | R-1 ↑ |
| MSMT17 (Base) | 63.5 | 84.0 | 75.7 | 86.2 |
| DG-Net (Zheng et al., 2019) | 60.8 | 82.1 | – | – |
| DPTN (Zhang et al., 2022) | 62.0 | 83.0 | – | – |
| Pose2ID (Yuan et al., 2025a) | 64.3 | 84.6 | – | – |
| OmniPerson (Ma et al., 2025) | 65.1 | 85.1 | – | – |
| **ReasonAug (Ours)** | **66.1** | **85.8** | **77.8** | **88.0** |

*Table 3.* Impact of Instruction Policy on Market-1501 with the **same frozen generator**. ReasonAug outperforms powerful generalist VLMs (GPT-4V) and imitation learning (SFT), confirming the value of task-specific RL planning.

| Policy Source | Strategy | Div. ↑ | mAP ↑ | Δ |
|---|---|---|---|---|
| A. Random | Random Attributes | 0.38 | 87.5 | – |
| B. GPT-4V | Zero-shot Captioning | 0.35 | 87.9 | +0.4 |
| C. Rule-Based | Heuristic Rules | 0.32 | 88.0 | +0.5 |
| D. SRA (SFT) | Imitation Learning | 0.40 | 88.1 | +0.6 |
| E. **ReasonAug** | **RL Planning** | **0.45** | **89.0** | **+1.5** |

## 4.2. Main Results: Data Augmentation Performance

We evaluate the effectiveness of ReasonAug as a data augmentation tool. Following the protocol in OmniPerson (Ma et al., 2025), we combine the generated samples with the original training set and train two representative ReID models: **TransReID**(He et al., 2021): A Transformer-based baseline. **CLIP-ReID**(Li et al., 2023): A strong multimodal baseline.

**Results on Market-1501.** Table 1 compares ReasonAug with recent generative augmentation methods. We observe two consistent trends. (i) Compared with end-to-end generator training (OmniPerson, 88.7% mAP with TransReID), ReasonAug reaches **89.0% mAP** with a *frozen* editor, indicating that instruction planning remains a major bottleneck once the editor is sufficiently capable. (ii) Gains transfer across backbones: on CLIP-ReID, ReasonAug attains **90.6%** mAP, slightly surpassing OmniPerson, suggesting the learned instruction policy is not backbone-specific (see Fig. 4 for training dynamics). **Stronger backbones.** Using the same generated data without policy retraining, ReasonAug also improves stronger models: SOLIDER (Swin-B) increases from 91.0/95.8 to 91.7/96.1 mAP/R-1, and CLIP-ReID (ViT-L/14) increases from 92.3/96.2 to 92.8/96.5. Since ReasonAug modifies the training data rather than the loss or architecture, it is orthogonal to centroid-based ReID objectives such as CTL/MCTL.

**Results on MSMT17.** We further evaluate on the more challenging MSMT17 dataset (Table 2). MSMT17 contains complex lighting and background variations, making it an ideal testbed for our Semantic Reasoning Agent. ReasonAug achieves significant gains (+2.6% mAP over baseline and +1.0 over OmniPerson), demonstrating that our SAE mechanism effectively explores diverse environmental factors (e.g., lighting, weather) that are critical for MSMT17 but sparse in the original distribution.

## 4.3. The "Instruction Gap": Policy Matters

To validate our core hypothesis—that the bottleneck lies in *what to generate* rather than *how to generate*—we conduct a controlled experiment. We keep the generator (Frozen MMDiT + TAR) constant and vary the source of instructions.

**Closed-loop feedback vs. stronger prompting.** We further isolate whether the gain comes from RL optimization rather than larger base-model capability. Under the same Qwen3-VL-2B backbone, open-loop few-shot prompting reaches only $83.0 \pm 0.7$ mAP, while the closed-loop RL policy reaches 89.0 mAP. A stronger open-loop Qwen3-VL-4B-thinking model reaches $84.8 \pm 0.7$ mAP, still below the RL-tuned 2B policy. Thus, downstream ReID feedback provides information that prompt engineering alone does not access.

**Generator strength.** Our claim is not that generator quality is irrelevant, but that instruction planning remains a major bottleneck after the editor reaches reasonable quality. With the learned policy fixed, stronger editors improve generation quality and mAP moderately: SD-v1.5+ControlNet (88.2), SDXL+IP-Adapter (88.6), FLUX.1 Kontext at 4/28 steps

*Table 4.* Ablation of Reward Design. **Hardness** indicates the discriminative difficulty of generated samples. Naive Triplet reward leads to hacking (high hardness but low ID score). MAGR achieves the best balance.

| Reward Signal | ID Score | Hardness | mAP |
|---|---|---|---|
| ReID Loss (Softmax) | **0.93** | 0.42 | 88.2 |
| Triplet (No Gate) | 0.72 | **0.88** | 86.4 |
| **MAGR (Ours)** | 0.91 | 0.65 | **89.0** |

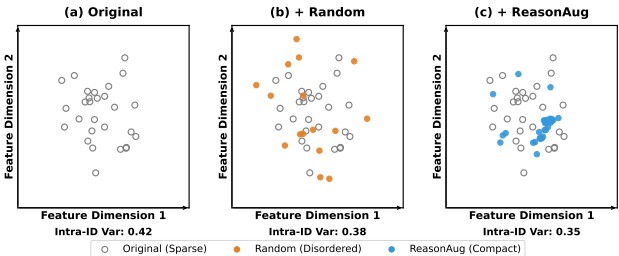

*Figure 3.* **Feature distribution (t-SNE).** (a) Original samples show sparse regions. (b) Random augmentation fails to fill gaps. (c) ReasonAug targets under-represented regions, reducing intra-class variance ($0.42 \rightarrow 0.35$).

(88.5/89.0), and ChronoEdit-14B (89.1). This variation is smaller than the 1.5 mAP gap between random instructions and ReasonAug under the same FLUX editor.

### 4.4. Mechanism Validation: Why RL Works?

We perform ablation studies to validate the two core components designed for the RL stage: the Metric-Aligned Gated Reward (MAGR) and Structure-Aware Entropy (SAE).

**1. Reward Design: The Necessity of MAGR.** Standard RL often suffers from reward hacking. We compare MAGR against standard ReID classification loss (Softmax) and a naive Triplet reward without identity gating. We define **Hardness** as the average margin violation, computed as $\mathbb{E}[d(a, p) - d(a, n)]$, where larger values indicate more difficult triplets.

As shown in Table 4, using only ReID classification loss results in conservative generation (Low Hardness), limiting performance gains. Conversely, removing the identity gate (Naive Triplet) causes *reward hacking*: the agent generates adversarial noise to maximize distance, destroying identity (ID Score 0.72). MAGR successfully guides the agent to the "sweet spot"—generating valid (ID 0.91) yet challenging (Hardness 0.65) samples. Fig. 3 further visualizes this effect: ReasonAug-generated samples fill sparse regions in the feature space, yielding a more compact intra-class distribution.

**2. Entropy Control: Resolving the Paradox.** We verify

*Table 5.* Ablation of Entropy Strategy. Uniform regularization forces a trade-off between ID preservation and diversity. SAE (Dynamic) achieves high diversity without sacrificing identity.

| Entropy Mode | ID Score | Div. ↑ | mAP |
|---|---|---|---|
| Uniform ($\lambda = 0.05$) | **0.94** | 0.28 | 88.3 |
| Uniform ($\lambda = 0.20$) | 0.81 | 0.36 | 87.9 |
| **SAE (Dynamic)** | 0.91 | **0.45** | **89.0** |

*Table 6.* Ablation of Chain-of-Thought. CoT improves instruction quality by identifying discriminative attributes before generation.

| Reasoning Mode | ID Score | Attr. Acc. | mAP |
|---|---|---|---|
| Direct Instruction | 0.88 | 0.72 | 88.4 |
| **With CoT (Ours)** | **0.91** | **0.85** | **89.0** |

whether SAE successfully resolves the stability-diversity dilemma compared to uniform entropy regularization.

Table 5 confirms the paradox: low uniform entropy leads to mode collapse (Div. 0.28), while high uniform entropy causes identity drift (ID Score 0.81). We swept $\lambda \in \{0.01, 0.05, 0.1, 0.2\}$ and observed a consistent trade-off; SAE shifts the Pareto frontier. SAE dynamically allocates entropy based on token sensitivity, achieving the highest diversity (Div. 0.45) while maintaining strong identity consistency. We also compared against simpler token-level alternatives: no entropy (88.0 mAP), random token weighting (88.2), manual POS-category weighting (88.5), and attention-only weighting without outcome conditioning (88.6). SAE performs best because it combines visual grounding with rollout outcomes. To test proxy robustness, we injected Gaussian noise into the cross-attention scores; even with 20% noise (Spearman correlation drops from 0.72 to 0.58), SAE retains 88.7 mAP, still above uniform entropy.

**3. Chain-of-Thought: Does Reasoning Help?** A key design choice of SRA is to generate a Chain-of-Thought (CoT) analysis before outputting the final instruction. We ablate this by comparing against a variant that directly outputs instructions without explicit reasoning.

As shown in Table 6, CoT reasoning improves both identity preservation (ID Score $0.88 \rightarrow 0.91$) and attribute accuracy (Attr. Acc. $0.72 \rightarrow 0.85$). We define **Attr. Acc.** as the percentage of identity-critical attributes (e.g., backpack straps, logos) correctly preserved in generated images. The explicit reasoning step helps the agent identify *which* attributes are discriminative before deciding *how* to vary them, leading to more targeted instructions.

**4. SAE under High-Intensity Augmentation.** A critical concern is whether the policy maintains identity under aggressive augmentation (e.g., extreme pose or background

*Table 7.* SAE robustness under high-intensity augmentation. SAE maintains stable identity even when diversity pressure increases, while uniform entropy suffers from identity collapse.

| Augmentation Intensity | Method | ID Score | Div. |
|---|---|---|---|
| Normal ($\beta = 0.1$) | Uniform | 0.89 | 0.32 |
| | SAE | **0.91** | **0.45** |
| High ($\beta = 0.3$) | Uniform | 0.71 | 0.48 |
| | SAE | **0.88** | **0.52** |
| Extreme ($\beta = 0.5$) | Uniform | 0.58 | 0.55 |
| | SAE | **0.84** | **0.58** |

*Table 8.* Sample Efficiency. ReasonAug achieves saturation faster, requiring 4× fewer samples to match the performance of Random Policy.

| Samples / ID | 1 | 2 | 4 | 8 |
|---|---|---|---|---|
| Random Policy | 87.5 | 87.8 | 88.0 | 88.1 |
| **ReasonAug** | **88.3** | **88.7** | **89.0** | **89.2** |

changes). We stress-test SAE by increasing the diversity pressure and measuring identity drift.

Table 7 shows that under high-intensity augmentation ($\beta = 0.3, 0.5$), uniform entropy suffers severe identity collapse (ID Score drops to 0.71, 0.58), while SAE maintains robust identity preservation (0.88, 0.84). This confirms that SAE's token-level entropy allocation effectively "locks" identity-critical tokens even under aggressive diversity pressure, directly addressing concerns about identity drift.

### 4.5. Efficiency Analysis

Finally, we demonstrate the practical value of ReasonAug in terms of data efficiency.

**Sample Efficiency.** Does "better planning" translate to "less data"? We compare the downstream mAP gain as a function of the number of generated images per identity.

With just **1 image/ID**, ReasonAug (88.3%) outperforms the Random Policy with 8 images/ID (88.1%). This implies that our RL agent prioritizes the most informative variations (e.g., rare poses), avoiding redundant sampling. The one-time offline cost is about 66 hours on 8×H800 GPUs (6h SFT + 60h RL); after data generation, downstream ReID training has no additional optimization overhead beyond using the augmented images.

### 4.6. Long-tail and Instruction Quality Analysis

ReasonAug mitigates long-tail bias by learning *what to generate* rather than sampling attributes uniformly. On Market-1501, the attribute entropy of generated instructions

increases from 3.08 bits (original) to 3.39 bits (ReasonAug), and improvements are substantially larger on rare categories (e.g., side-view/occluded) than on common categories. We also evaluate instruction executability. Using a VLM-as-a-judge protocol (500 samples), ReasonAug achieves an instruction-following score of 8.3±0.7, outperforming SFT-only (6.9±1.0); RL further shifts the policy toward spatially-localized vocabulary (spatial terms increase from 20% to 50%), reducing generator ambiguity. Detailed distributions, breakdown tables, and analyses are provided in Appendix A (Figs. 5, 6, 7, 8; Table 9).

**Additional dataset.** On the more fragile CUHK03-Detected benchmark with CLIP-ReID, ReasonAug improves from 72.5/74.3 to 76.5/78.2 mAP/R-1, exceeding random augmentation (73.1/75.0). This suggests that closed-loop instruction learning is especially useful when identity cues are sparse or noisy.

### 4.7. Occluded-ReID Specialized Testing

Occlusion is a critical challenge in real-world ReID. We verify that ReasonAug improves robustness on occluded benchmarks and that the policy learns to generate occlusion-aware instructions; full results and analysis are deferred to Appendix A (Table 10 and Fig. 9).

### 4.8. Training Dynamics: MAGR-Induced Curriculum and SAE-Induced Entropy Separation

Figure 4 summarizes how ReasonAug evolves during RLVR.

**1. The Gating Mechanism Induces a Natural Curriculum.** We set the gate threshold $\delta = 0.5$ for all experiments. As shown in Figure 4(a) and (b), the training process exhibits three distinct stages defined by the gate activation rate $\Pr(R_{id} > \delta)$: **Stage A (Warm-up, $\Pr(R_{id} > \delta) < 0.5$; steps 0–1200):** updates are dominated by $R_{id}$ since $R_{task}$ is effectively masked by the gate (and can be negative when mined triplets violate the margin). **Stage B (Switch-on, $0.5 \leq \Pr(R_{id} > \delta) < 0.8$; steps 1200–3500):** once the gate activates more frequently, the effective contribution of $R_{task}$ increases and we observe a sharp rise of $R_{task}$. **Stage C (Refinement, $\Pr(R_{id} > \delta) \geq 0.8$; steps 3500+):** both rewards stabilize at high levels, achieving fidelity and hardness. This confirms that MAGR effectively prevents reward hacking: the agent is not allowed to optimize metric distance until it respects the identity manifold.

**2. SAE Successfully Disentangles Entropy.** Figure 4(c) plots the entropy gap $\Delta H = H_{var} - H_{id}$ between variation-related tokens and identity-critical tokens. We compute $H_{id}$ and $H_{var}$ by grouping tokens using the same attention-based mask used by SAE. Initially, the gap is small ($\sim 0.1$), as the SFT model treats all tokens similarly. As training pro-

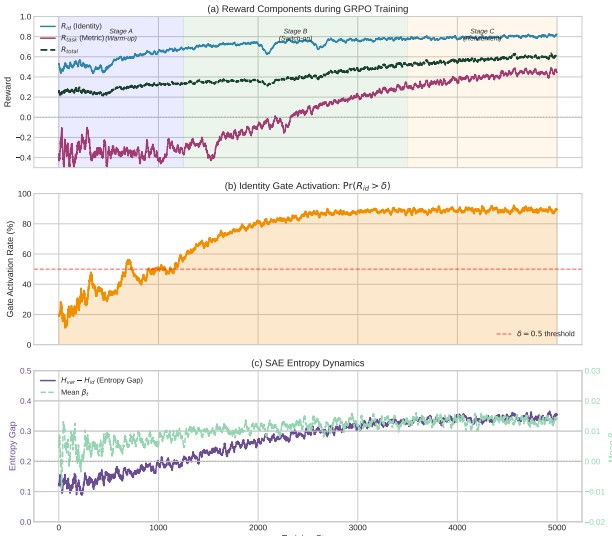

*Figure 4.* **Training Dynamics of ReasonAug. (a)** Evolution of reward components. A natural curriculum emerges: the agent first focuses on Identity Preservation ($R_{id}$, Stage A), and only after the Identity Gate opens consistently **(b)**, does it optimize for Metric Discriminability ($R_{task}$, Stage B & C). **(c)** SAE Dynamics. The widening gap between variation entropy ($H_{var}$) and identity entropy ($H_{id}$) confirms that the agent learns to "lock" identity-critical attributes (low entropy) while increasing diversity in other regions (high entropy).

gresses under SAE, the gap widens significantly to $\sim 0.35$. Concretely, the learned policy becomes **low-entropy on identity tokens** (deterministic clothing/accessories) while remaining **high-entropy on variation tokens** (diverse backgrounds/poses). This emergent divergence provides strong empirical evidence that our structure-aware entropy mechanism works as intended, resolving the stability-diversity dilemma.

## 5. Conclusion and Limitations

We presented **ReasonAug**, which reformulates generative ReID augmentation as **instruction policy learning** for a **frozen** image editor. Our key insight is that with strong modern generators, the bottleneck shifts from *how to generate* to *what to generate*. By training a Semantic Reasoning Agent via RLVR, ReasonAug achieves state-of-the-art augmentation gains by unlocking the planning capabilities of VLM without the need for expensive generator fine-tuning.

We further introduce **MAGR** and **SAE** to align rewards with metric discriminability and stabilize exploration, enabling diverse yet identity-preserving edits that outperform heuristic prompting.

**Limitations.** Our method relies on the quality of the ReID reward model and may inherit its biases, and RLVR remains computationally expensive. A manual audit of 500 genera-

tions shows three typical failure modes: unrealistic artifacts under extreme poses (4.4%), attribute confusion in multi-person scenes (7.3%), and fine-grained texture loss such as blurred logos (12.0%). The identity gate suppresses the first two modes because they receive low identity-preservation scores, but very fine textures remain limited by the frozen generator resolution. Future work includes stronger reward ensembles, higher-resolution editors, and more efficient RL optimization.

## Impact Statement

This paper presents work whose goal is to advance the field of machine learning through closed-loop generative data augmentation for person re-identification (ReID). If applied responsibly, the proposed framework may improve robustness and generalization of ReID models and reduce reliance on additional data collection by better leveraging existing datasets.

At the same time, ReID technologies can be used in surveillance settings and may raise privacy concerns. Moreover, generative augmentation may amplify dataset biases (e.g., demographic imbalance) or be misused to create deceptive imagery. We emphasize that our work is intended for research purposes and recommend that any deployment follow applicable laws and institutional policies, include privacy protections and access controls, and be accompanied by bias and performance auditing across relevant subpopulations.

## Acknowledgements

This work was partially supported by the National Natural Science Foundation of China (No. 62472016) and the Fundamental Research Funds for the Central Universities.

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

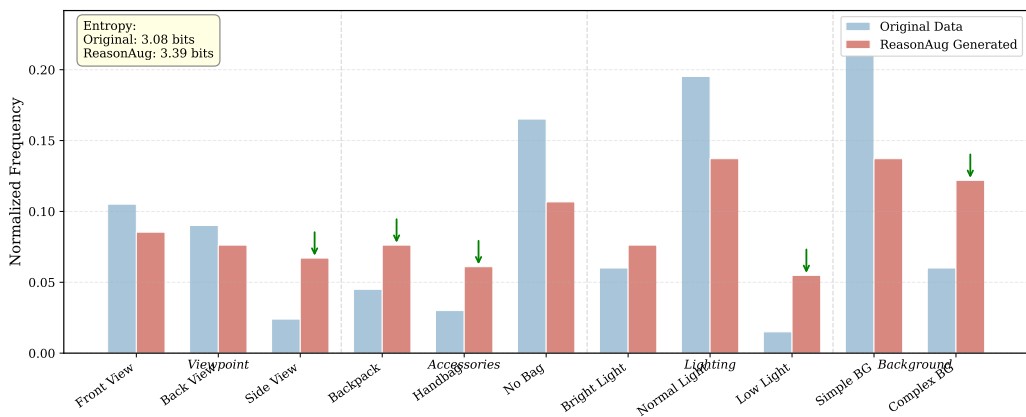

*Figure 5.* **Attribute frequency distribution.** Original data (blue) shows long-tail distribution; ReasonAug (red) generates more balanced coverage, particularly for rare attributes (side view, accessories, complex backgrounds). Entropy increases from 3.08 to 3.39 bits.

*Table 9.* Performance on common vs. rare categories (Market-1501). ReasonAug shows significantly larger improvements on rare categories.

| Category | Baseline | ReasonAug | Δ |
|---|---|---|---|
| Front View (Common) | 88.5 | 89.2 | +0.7 |
| Back View (Common) | 87.2 | 88.1 | +0.9 |
| Side View (Rare) | 72.3 | 78.6 | **+6.3** |
| Occluded (Rare) | 68.5 | 75.2 | **+6.7** |

## A. Additional Experimental Analyses

### A.1. Long-tail Mitigation Analysis Details

**Attribute Entropy Analysis.** We compare the frequency distribution of semantic attributes (viewpoint, accessories, lighting, background) between the original Market-1501 training set and ReasonAug-generated instructions. The original data exhibits a skewed (long-tail) distribution—front/back views dominate while side views are rare; most subjects have no bag; complex backgrounds are underrepresented. In contrast, ReasonAug generates instructions that yield a more balanced distribution, with higher frequencies on originally rare attributes. The information entropy increases from 3.08 bits (original) to 3.39 bits (ReasonAug), confirming that our RL policy actively explores under-represented semantic regions.

**Performance on Tail Categories.** We partition the Market-1501 test set into common categories (Front/Back view) and rare categories (Side view, Occluded). Table 9 shows that ReasonAug yields modest gains on common categories (+0.7%–0.9% mAP) but **significantly larger improvements on rare categories (+6.3%–6.7% mAP)**.

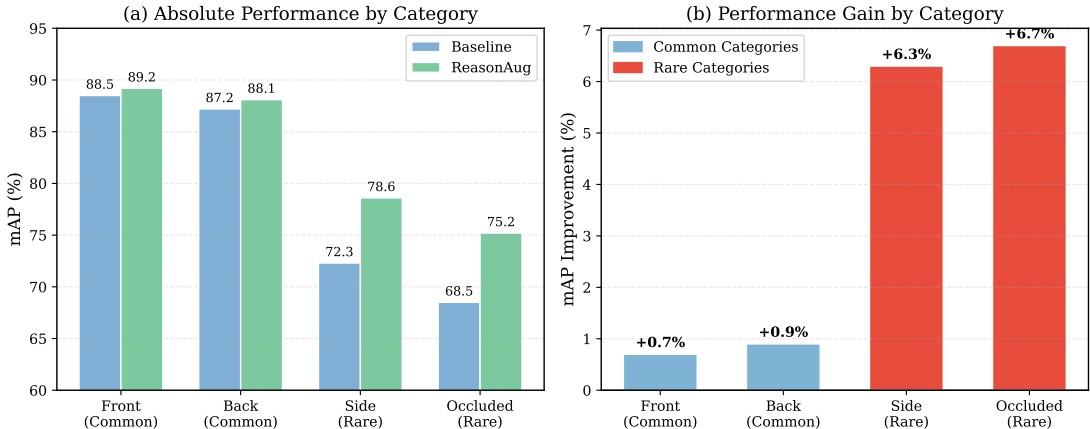

*Figure 6.* **Tail category performance analysis.** (a) Absolute mAP by category. (b) Performance gain: rare categories (red) show 6–7× larger improvements than common categories (blue).

## A.2. Instruction Following Score and Vocabulary Evolution

**VLM-as-a-Judge Evaluation.** We randomly sample 500 (instruction, generated image) pairs and use GPT-4o as an impartial judge to score instruction following on a 1–10 scale.

**Vocabulary Evolution.** RL training induces a shift from generic to spatially-localized vocabulary. After RL, the agent uses fewer generic terms ("bag", "walking") and more spatial descriptors ("left shoulder", "back pocket"). The composition shifts from 65% generic / 20% spatial (SFT) to 30% generic / 50% spatial (RL).

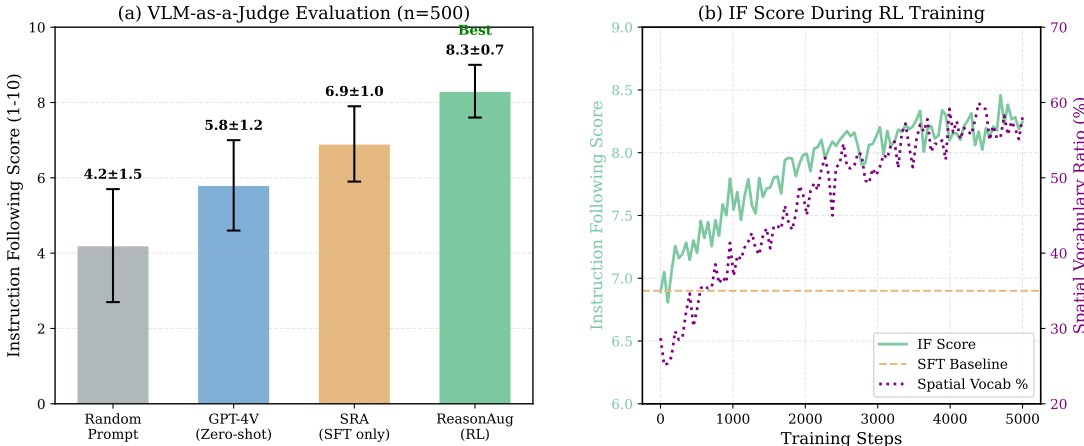

*Figure 7.* **Instruction Following Score.** (a) VLM-as-a-Judge evaluation shows RL significantly improves instruction quality. (b) Training dynamics: IF score increases alongside spatial vocabulary ratio.

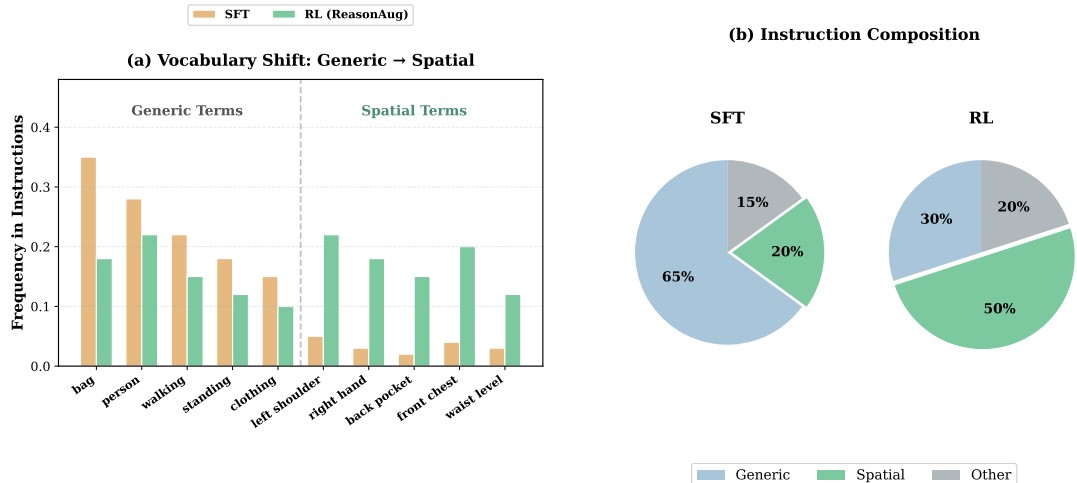

*Figure 8.* **Vocabulary evolution during RL.** (a) Frequency shift from generic to spatial terms. (b) Instruction composition: spatial vocabulary increases from 20% to 50%.

### A.3. Occluded-ReID Specialized Testing

**Results on Occluded Benchmarks.** Table 10 shows results on three occluded ReID datasets.

**Occlusion Instruction Analysis.** After RL, SRA generates significantly more occlusion instructions: pole occlusion (5%→18%), person occlusion (8%→25%), partial view (15%→28%).

*Table 10.* Performance on occluded ReID benchmarks. ReasonAug learns to generate occlusion-aware augmentations.

| Method | Occ-Duke | Occ-ReID | Partial-ReID |
|---|---|---|---|
| Baseline | 42.3 | 58.5 | 61.2 |
| Random Aug | 44.1 | 60.2 | 62.8 |
| **ReasonAug** | **51.8** | **67.3** | **69.5** |
| $\Delta$ (vs Base) | **+9.5** | **+8.8** | **+8.3** |

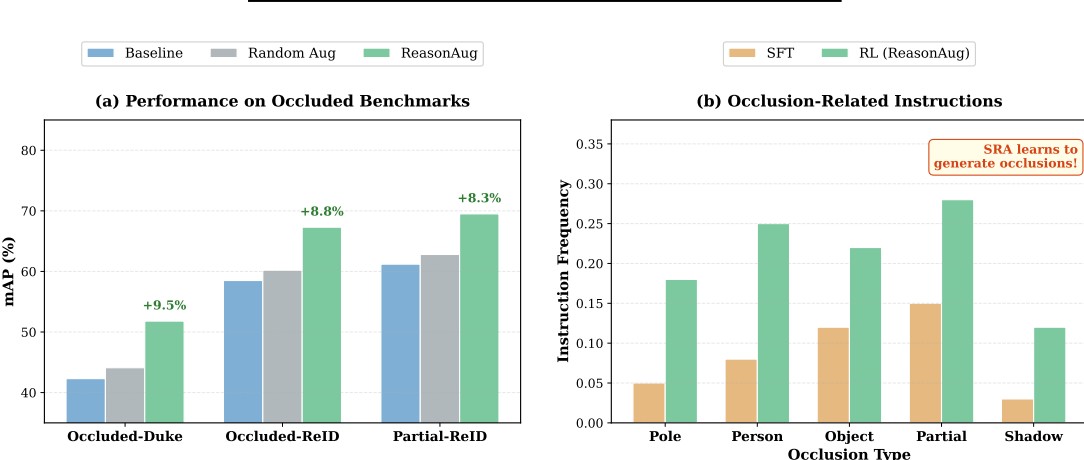

*Figure 9.* **Occluded-ReID analysis.** (a) Performance comparison on occluded benchmarks. (b) SRA learns to generate occlusion-related instructions during RL training.

# B. System Implementation Notes

### B.1. System Architecture

We implement ReasonAug using a distributed RLVR framework. The training pipeline consists of decoupled *Actors* (for generation), *Reward Models* (for feedback), and a *Learner* (for policy updates). This decoupling allows for scalable training across multiple GPUs.

### B.2. Efficient Token Sensitivity Extraction

To minimize computational overhead, SAE uses the same cross-attention proxy described in Sec. 3.4, rather than an additional reward-gradient attribution module. During actor-side generation, the frozen editor already produces cross-attention maps between instruction tokens and image regions. We pool these maps to obtain token sensitivity scores $S_t$ and threshold them into the binary mask $w_t$, which separates identity-critical tokens from lower-sensitivity variation tokens. The actor packs $\{S_t, w_t\}$ together with the generated image and reward into the trajectory data structure and sends them to the learner. Consequently, the Structure-Aware Entropy loss can be computed during policy updates without backpropagating through the ReID reward model or re-running the editor.

# C. Metric-Aligned Gated Reward (MAGR) Illustration

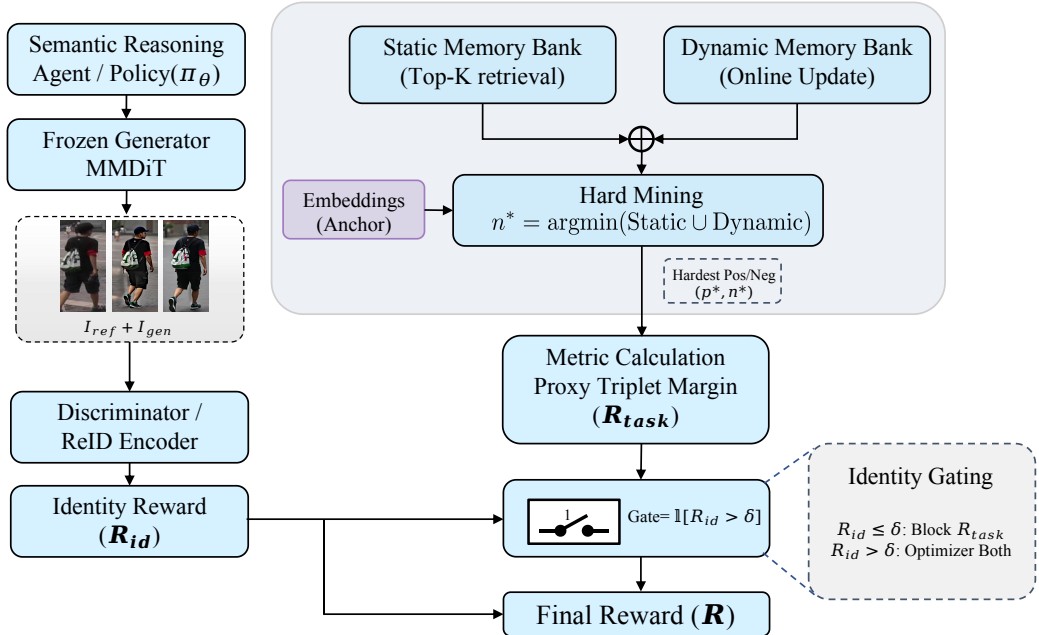

*Figure 10.* **Metric-Aligned Gated Reward (MAGR).** A two-layer memory bank (static prior + dynamic posterior) mines hard negatives $(n^*)$ and positives $(p^*)$ for distance-based rewards. An identity gate $(R_{id} > \delta)$ activates $R_{task}$ only when identity is preserved, preventing reward hacking.

## D. Cross-Attention Proxy Validation

We define the proxy token sensitivity $S_{\text{proxy}}$ as the pooled cross-attention response used in SAE:

$$S_{\text{proxy}}(t) = \text{Pool}_{\text{spatial,layers}}(\text{AttnMap}_t). \tag{14}$$

To validate that $S_{\text{proxy}}$ approximates an offline diagnostic sensitivity $S_{\text{ideal}}$, we conducted the following experiment on 100 randomly sampled images from Market-1501:

**Protocol.** For each image, we: (1) Generate a CoT output and corresponding image via our pipeline. (2) Compute $S_{\text{proxy}}$ using Eq. (14). (3) Approximate $S_{\text{ideal}}$ via finite differences: perturb each token embedding by $\epsilon = 10^{-3}$, regenerate the image, and compute $|R_{\text{id}}^{\text{pert}} - R_{\text{id}}^{\text{orig}}|/\epsilon$. (4) Compute Spearman rank correlation between the two scores.

**Results.** We observe $\rho = 0.72$ with $p < 0.01$, indicating a strong positive correlation. Figure 11 shows the scatter plot.

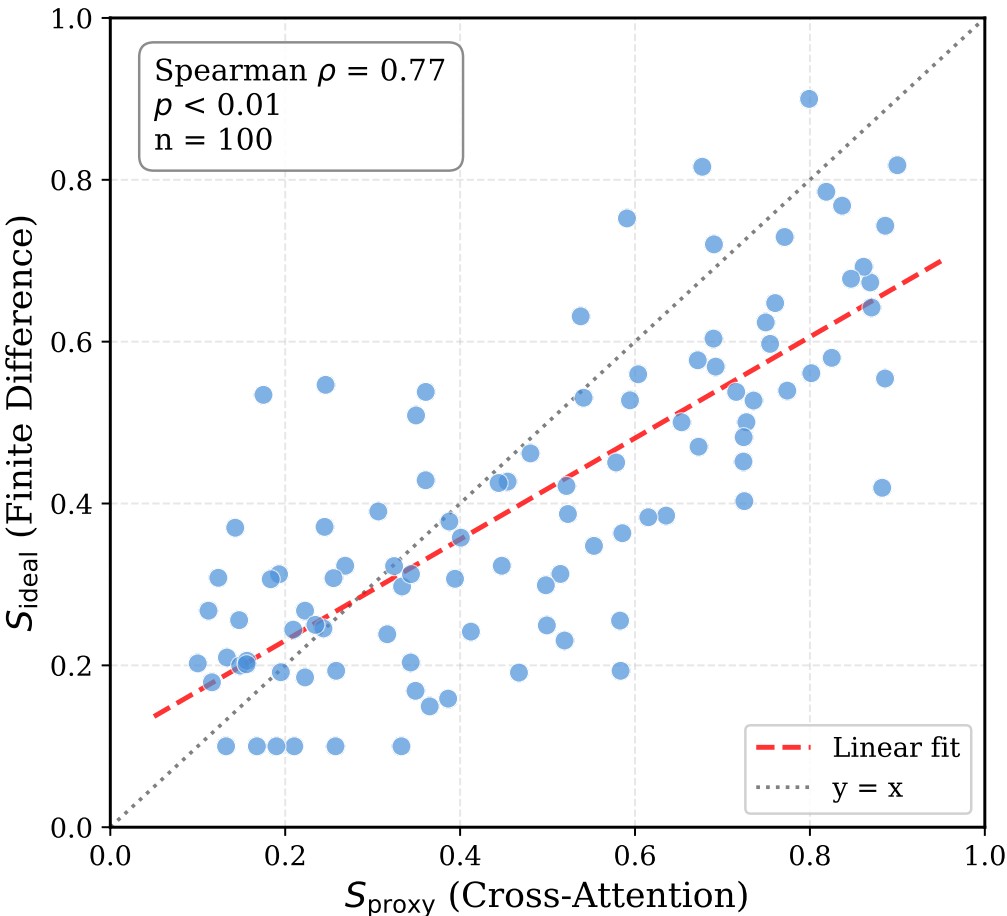

*Figure 11.* Scatter plot between $S_{\text{proxy}}$ (cross-attention based) and $S_{\text{ideal}}$ (finite difference approximation) for 100 randomly sampled images from Market-1501. Strong positive correlation ($\rho = 0.72$, $p < 0.01$) validates that cross-attention serves as an effective proxy for token importance in identity preservation.

## E. Gradient Dynamics of Structure-Aware Entropy

To provide a rigorous theoretical grounding for the Structure-Aware Entropy (SAE) mechanism, we derive the exact gradient of the entropy term with respect to the pre-softmax logits. This derivation reveals how the dynamic coefficient $\beta_t$ modulates the probability distribution shape during optimization.

### E.1. Derivation of the Entropy Gradient

Let the policy output at step $t$ be a categorical distribution over a vocabulary of size $V$. Let $\mathbf{z} \in \mathbb{R}^V$ denote the unnormalized logits, and $\mathbf{p} = \text{softmax}(\mathbf{z})$ be the probability distribution:

$$p_i = \frac{e^{z_i}}{\sum_{j=1}^{V} e^{z_j}} \tag{15}$$

The Shannon entropy of the distribution is:

$$H(\mathbf{p}) = -\sum_{i=1}^{V} p_i \log p_i \tag{16}$$

We compute the partial derivative $\frac{\partial H}{\partial z_k}$ via the chain rule:

$$\frac{\partial H}{\partial z_k} = \sum_{i=1}^{V} \frac{\partial H}{\partial p_i} \frac{\partial p_i}{\partial z_k} \tag{17}$$

The derivative of entropy with respect to probability is:

$$\frac{\partial H}{\partial p_i} = -(\log p_i + 1) \tag{18}$$

The softmax Jacobian is:

$$\frac{\partial p_i}{\partial z_k} = p_i(\delta_{ik} - p_k) \tag{19}$$

Substituting into the chain rule:

$$\frac{\partial H}{\partial z_k} = \sum_{i=1}^{V} -(\log p_i + 1) \cdot p_i(\delta_{ik} - p_k) \tag{20}$$

$$= -\sum_{i=1}^{V} p_i \log p_i(\delta_{ik} - p_k) - \underbrace{\sum_{i=1}^{V} p_i(\delta_{ik} - p_k)}_{=0} \tag{21}$$

The second term vanishes because $\sum_i \frac{\partial p_i}{\partial z_k} = \frac{\partial}{\partial z_k}(1) = 0$.

Expanding the remaining term:

$$\frac{\partial H}{\partial z_k} = - \left[ p_k \log p_k(1 - p_k) - p_k \sum_{i \neq k} p_i \log p_i \right] \tag{22}$$

Using the identity $\sum_{i \neq k} p_i \log p_i = -H - p_k \log p_k$:

$$\frac{\partial H}{\partial z_k} = - \left[ p_k \log p_k - p_k^2 \log p_k + p_k H + p_k^2 \log p_k \right] \tag{23}$$

$$= -p_k \left( \log p_k + H \right) \tag{24}$$

$$= p_k \left( -\log p_k - H \right) \tag{25}$$

**Intuitive Interpretation.** The term $-\log p_k$ represents the "surprise" of token $k$, while $H$ is the average surprise. The gradient is positive when $-\log p_k > H$ (i.e., $p_k < e^{-H}$), pushing low-probability tokens upward. Conversely, dominant tokens with $p_k > e^{-H}$ receive negative gradients, redistributing probability mass toward uniformity.

### E.2. Optimization Dynamics under SAE

In our framework, the loss includes a term $\mathcal{L}_{\text{SAE}} = -\beta_t H$. The gradient update on $z_k$ receives a contribution:

$$-\frac{\partial \mathcal{L}_{\text{SAE}}}{\partial z_k} = \beta_t \frac{\partial H}{\partial z_k} = \beta_t \cdot p_k(-\log p_k - H) \tag{26}$$

**Scenario 1: Identity-Critical Tokens ($\beta_t = \lambda_{low} \approx 0$).**
The entropy gradient contribution is negligible. Optimization is dominated by the reward signal $R_{id}$, which maximizes the log-probability of identity-preserving tokens. Without entropy regularization to "flatten" the distribution, the policy converges to a sharp, near-deterministic output. (The KL penalty provides a soft constraint preventing extreme sharpness.)

**Scenario 2: Variation Tokens ($\beta_t = \lambda_{high} > 0$).**
The gradient actively encourages entropy maximization. For a dominant token $k$ with $p_k > e^{-H}$, the term $\beta_t p_k(-\log p_k - H)$ is **negative**, pushing $z_k$ downward. This redistributes probability mass to alternative tokens (e.g., diverse poses or backgrounds), mathematically enforcing exploration and preventing mode collapse.

# F. Implementation Details and Reproducibility

For reproducibility, we provide comprehensive details of all components in our framework.

### F.1. Generator Details

We use a pruned version of **Flux.1 Kontext** (Labs et al., 2025) as our frozen instruction-following image editor. The VAE is from SDXL (Podell et al., 2023). Key inference parameters:

- Resolution: $512 \times 768$ (height $\times$ width)

- Inference steps: 28

- Guidance scale: 3.5

- Scheduler: Flow matching with shifted timesteps

### F.2. Text Alignment Refiner (TAR)

TAR bridges the gap between the VLM's detailed reasoning output and the generator's conditioning space.

- Architecture: 2-layer MLP with LayerNorm ($\sim$2.3M parameters)

- Training data: 30K image-text pairs from Market-1501 training set, with captions synthesized by Qwen3-VL (Bai et al., 2025a)

- Loss: Cross-entropy

- Training: 3 epochs, batch size 32, learning rate $1 \times 10^{-4}$

### F.3. Reward Model

- Architecture: **ResNet50-IBN** (Pan et al., 2018), pre-trained on Market-1501

- **Independence**: The reward encoder is distinct from downstream evaluation models (TransReID, CLIP-ReID), ensuring the policy learns generalizable visual semantics

- Embedding dimension: 2048 ($\ell_2$-normalized)

### F.4. RL Hyperparameters

Table 11 summarizes the key hyperparameters for GRPO training.

### F.5. Computational Resources

- Hardware: $8 \times$ NVIDIA H800 (80GB)

- SFT warm-start: $\sim$6 hours

- RL training: $\sim$60 hours

- **Note**: Training is a one-time cost. At inference, only the VLM + Generator are needed (no reward model or RL gradients), enabling efficient deployment.

### F.6. Generation Budget

For fair comparison, all methods generate **4 images per identity** without filtering or rejection sampling. Total generated images: $\sim$3K for Market-1501 (751 identities $\times$ 4).

*Table 11.* RL training hyperparameters.

| Hyperparameter | Value |
|---|---|
| *GRPO* | |
| Group size $G$ | 4 |
| Clip ratio $\epsilon$ | 0.2 |
| KL coefficient $\gamma$ | 0.01 |
| *MAGR* | |
| Identity gate threshold $\delta$ | 0.5 |
| Triplet margin $m$ | 0.3 |
| Temperature $\tau$ | 0.1 |
| $w_{id}$, $w_{task}$ | 1.0, 1.0 |
| *SAE* | |
| Sensitivity quantile $\kappa$ | 0.7 |
| $\beta_{lock}, \beta_{explore}, \beta_{neutral}$ | 0.001, 0.1, 0.01 |
| *Training* | |
| Batch size | 8 |
| Learning rate | $1 \times 10^{-6}$ |
| Total steps | 5000 |
| Optimizer | AdamW |

## G. Baseline Prompt Templates

For the policy source comparison in Table 3, we provide the exact prompts used for each baseline to ensure fair comparison.

### G.1. GPT-4V (Zero-shot CoT)

We use a carefully designed CoT prompt that mirrors the structure of our SRA:

```
GPT-4V Prompt Template

Analyze this pedestrian image step by step:
1.   Identify global attributes (pose, viewpoint, scene)
2.   List identity-critical features (clothing, accessories)
3.   Suggest which factors can be varied without changing identity
4.   Output a final editing instruction in the format:  "Edit the image to [specific
change]"
Be specific about spatial locations (e.g., "on the left shoulder" rather than
"carrying a bag").
```

**Key observation**: Despite using an equivalent CoT structure, GPT-4V achieves lower performance than ReasonAug (Table 3). This confirms that the advantage of ReasonAug comes from **closed-loop feedback**—GPT-4V cannot observe whether its suggestions actually benefit the downstream ReID model, while our policy is optimized with verifiable rewards.

### G.2. Rule-Based Heuristics

We define 12 augmentation templates covering viewpoint, lighting, and background:

- Viewpoint: {`front, back, left side, right side view`}

- Lighting: {`bright sunlight, overcast, evening, indoor`}

- Background: {`street, mall, parking lot, campus`}

For each image, we randomly sample one template from each category with equal probability.

## Original    Generated

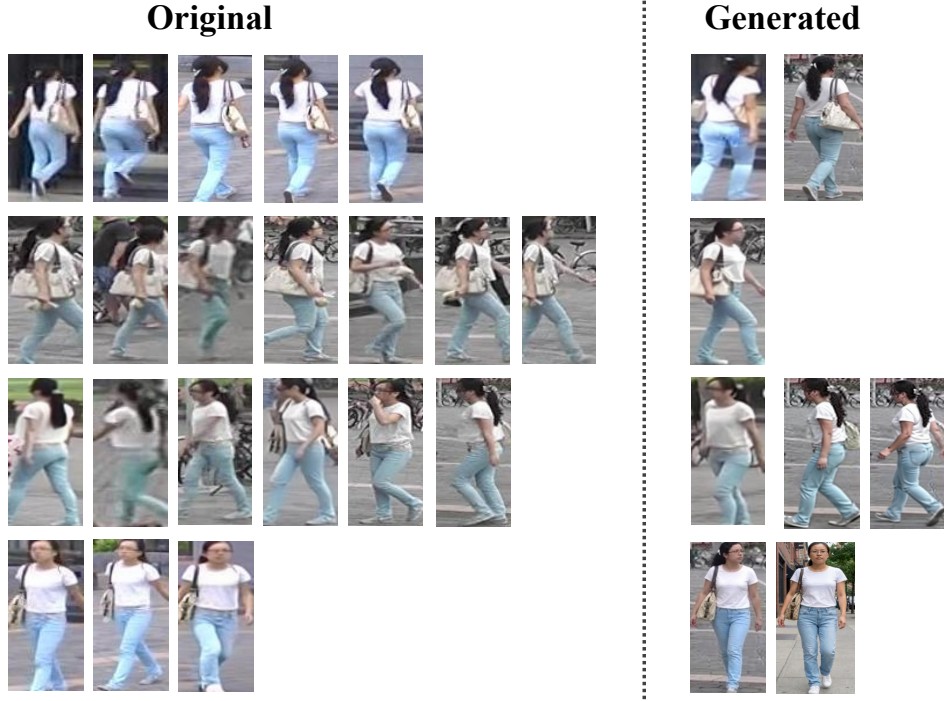

*Figure 12.* **Case Study 1: Filling the Viewpoint Vacuum.** For an identity with severe viewpoint bias in the original Market-1501 training set, ReasonAug synthesizes missing frontal views while preserving identity-critical attributes.

### G.3. Random Attribute Sampling

We randomly combine attributes from a predefined vocabulary (50 viewpoint descriptors, 30 lighting conditions, 40 background types) without any structured reasoning.

## H. Qualitative Case Studies

### H.1. Case Study 1: Filling the Viewpoint Vacuum

In Fig. 12, we showcase an identity from the Market-1501 dataset characterized by a severe viewpoint bias in the original training set. As seen in the left panel, the individual is predominantly captured from the side or back, with zero clear frontal views available.

ReasonAug autonomously recognizes this *feature vacuum*. The reasoning agent (SRA) plans an instruction to synthesize the missing frontal perspective. The resulting images (bottom right) provide a high-fidelity frontal view, preserving all identity-critical attributes: the specific beige tote bag with black straps, the long ponytail, and the light blue denim texture.

By synthesizing these unseen perspectives, ReasonAug expands the identity manifold, allowing the downstream ReID model to learn a holistic 360-degree representation of the individual, which is essential for matching targets across disparate camera views in real-world scenarios.

### H.2. Case Study 2: Action–Viewpoint Synergy (The Cycling Example)

As highlighted in the last row of Fig. 13, the reference identity is captured performing a rare action (cycling) only from a blurry side view in the original dataset. Standard augmentation methods would struggle to generalize this specific identity–action pair.

In contrast, our closed-loop agent (SRA) autonomously identifies this *semantic gap*. It plans and executes instructions to synthesize the same individual cycling from entirely new perspectives (front and back views), which are absent in the source distribution. This demonstrates that ReasonAug does not merely shuffle existing attributes, but actively extrapolates

**Original** | **Generated**

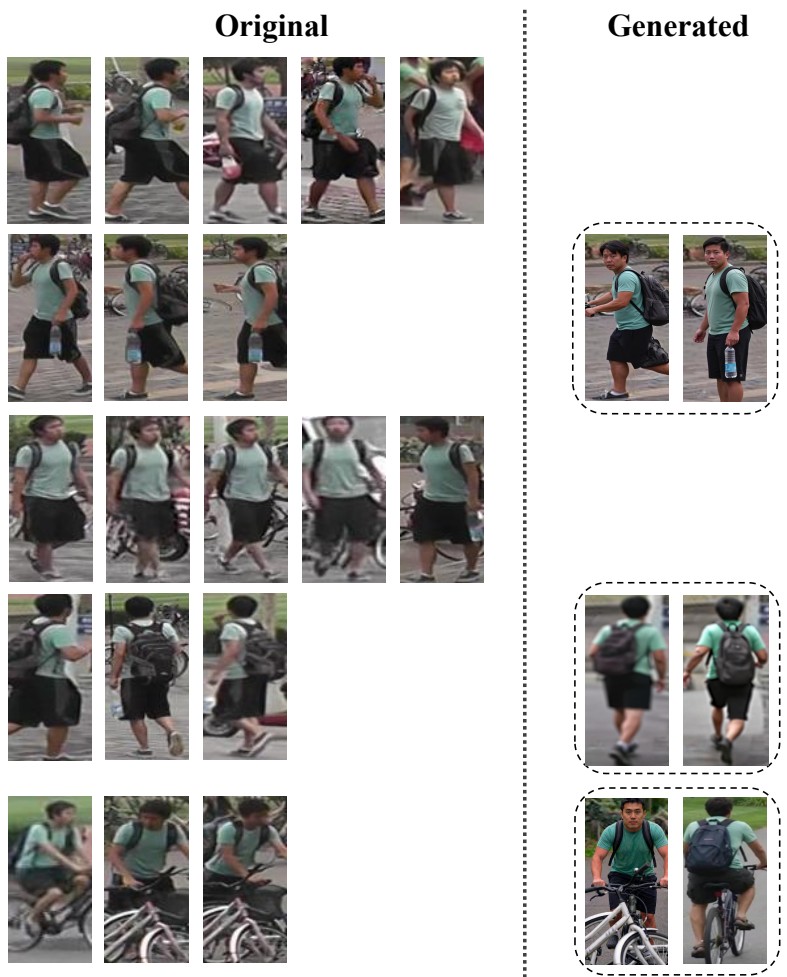

*Figure 13.* **Case Study 2: Action–Viewpoint Synergy.** For a rare identity–action pair (cycling) observed from a single blurry side view, ReasonAug synthesizes the same individual cycling from new viewpoints (front/back), enriching under-represented action–viewpoint subspaces.

the identity manifold into under-represented action–viewpoint subspaces, providing the downstream model with highly informative and discriminative training signals.

