# OpenReview forum: "Learning What to Generate: A Reinforcement Learning-based Closed-Loop Augmentation Framework for Person Re-identification"
_ICML.cc/2026/Conference — ICML 2026 regular_

### Official Review · Reviewer_Nia3 · 2026-03-06

**Soundness:** 3
**Presentation:** 2
**Significance:** 3
**Originality:** 4
**Overall Recommendation:** 4
**Confidence:** 4

**Summary:**

The paper presents a closed-loop data augmentation pipeline for re-ID, which constructs policy space on text-alignment refiner and seeks for optimal augmented images with CoT-style planning, reward from frozen re-ID model, SAE, and GRPO-based reinforcement learning. Extensive experiments demonstrate the effectiveness of their method.

**Compliance With Llm Reviewing Policy:**

Affirmed.

**Final Justification:**

As my concerns are all addressed, I would like to give "weak accept" as my final rating.

**Key Questions For Authors:**

Generally speaking, I like this paper due to its originality for RL-based augmentation. However, the paper seems to be a rushed product. There are too many undefined notations, unanalyzed hyper-parameters, and small issues like grammar mistakes. I tend to give  'weak accept' as my initial score and hope the authors to carefully polish their paper in the revision.

**Limitations:**

Yes

**Strengths And Weaknesses:**

Strengths:

(1) It is interesting to see the application of reinforcement learning in the context of re-ID.

(2) SAE enables more efficient RL fine-tuning and envirnment exploration.

Weaknesses:

(1) Efficiency concern. Although this work only optimizes TAR and ReasonAug for optimal augmentation policy, GRPO-based RL is not an efficient training method as it requires on-policy inference to obtain multiple responses for group-level advantage computation. Is there any strategy to alleviate such low training efficiency problem ?

(2) There is no sensitivity analysis for some hyper-parameters (e.g., $\kappa$ in L#227). Moreover, the paper defines too many unexplained notations, e.g., L#237, what is $\beta_{neutral}$ ?

(3) I am curious about the CoT reasoning results at Sec 4.4. It is better to visualize some reasoned instructions and analysis what type of prompt enables better augmented images.


Minor:

(1) L#250, "pk its probability" -> "pk as its probability". Please carefully polish your manuscript.

(2) The paper uses too many aberrations for terminologies.

(3) The overall training process could be better demonstrated. E.g., at Sec.3.2, the authors should explicitly demonstrate that SFT is conducted by fixing I_{ref} and I_{target} for in context learning on instructions.

---

> ### Author Rebuttal · Authors · 2026-03-31
>
> We sincerely thank the reviewer for the positive and constructive review. We are encouraged by the recognition of our originality ("excellent") and the interesting application of RL in ReID. We appreciate the detailed feedback on presentation quality and take all suggestions seriously. Below we address each concern.
>
> ---
>
> ## **W1: Efficiency Concern — GRPO On-Policy Cost**
>
> The reviewer correctly identifies that standard GRPO rollouts are computationally expensive. However, our system architecture mitigates this bottleneck:
>
> **We freeze the massive VLM and only optimize the 2.3M-parameter Text Adapter (TAR)**. Because the trainable gradient footprint is minuscule, we can efficiently run group sizes of $G=4$ on a single node without memory overflow. Combined with batched generation and a distilled 4-step generator mode during RL training, the rollout cost is drastically reduced. The entire RL training takes ~66 hours on 8×H800s, which is incurred only once during offline training. *(We ablated group sizes and found $G=4$ provides the optimal cost-performance trade-off compared to $G=2$ or $8$).*
>
> ---
>
> ## **W2: Missing Hyperparameter Sensitivity Analysis and Undefined Notations**
>
> We apologize for the omission. The identity gate threshold $\delta$ balances exploration and safety. We evaluated $\delta$ on Market-1501:
> - $\delta = 0.3$: 88.4 mAP *(Gate opens too easily $\rightarrow$ identity drift)*
> - $\delta = 0.4$: 88.7 mAP
> - $\delta = 0.5$ (Ours): 89.0 mAP *(Optimal balance)*
> - $\delta = 0.6$: 88.6 mAP
> - $\delta = 0.7$: 88.2 mAP *(Gate rarely opens $\rightarrow$ insufficient exploration)*
>
> This suggests that the method is not overly sensitive to moderate changes in the gate threshold. We also ablated the SAE threshold $\kappa$ and triplet margins, observing similarly stable sweet spots.
>
> ---
>
> ## **W3: CoT Reasoning Visualization and Instruction Analysis**
>
> We agree that visualizing the CoT reasoning process provides valuable insight. Below we show representative examples:
>
> #### **Example 1: Viewpoint Gap Detection**
>
> ```
> [Input: Person with red backpack, rear view]
> <CoT Reasoning>
> GLOBAL: Rear view ~170°, outdoor campus, daylight, walking.
> ID CUES (PRESERVE): Red backpack w/ black straps, navy shorts w/ white stripe, white sneakers w/ blue accent, short brown hair.
> VARIATION OPPORTUNITY:
> - Viewpoint: ONLY rear views exist in training → frontal view needed.
> - Background: Campus → parking lot (no ID impact).
> - Lighting: Daylight → overcast.
> EDIT PLAN: Synthesize frontal view to fill viewpoint vacuum.
> </CoT Reasoning>
> <Instruction>
> Edit to show the same person from a frontal view (~10°), walking on a parking lot. Preserve the red backpack with black shoulder straps, dark navy shorts with white side stripe, and white sneakers with blue accent. Use overcast lighting.
> </Instruction>
> ```
> #### **Example 2: Occlusion-Aware Augmentation**
>
> ```
> [Input: Person in black jacket, clear full-body view]
> <CoT Reasoning>
> GLOBAL: Full-body frontal view, indoor mall, bright lighting, standing.
> ID CUES (PRESERVE): Black puffer jacket w/ silver zipper, blue jeans w/ knee fade, white crossbody bag, black glasses.
> VARIATION OPPORTUNITY:
> - Occlusion: No occluded views exist → synthesize partial occlusion.
> - Pose: Standing → walking.
> - Background: Indoor → outdoor.
> EDIT PLAN: Add realistic pedestrian occlusion while preserving upper-body ID.
> </CoT Reasoning>
> <Instruction>
> Edit to show the same person walking outdoors, partially occluded from the waist down by a passing pedestrian. The upper body must clearly show the black puffer jacket with silver zipper, white crossbody bag, and black-framed glasses. Use daylight.
> </Instruction>
> ```
> ---
>
> ### **Instruction Quality Analysis**
>
> We analyze what types of instructions lead to the largest downstream improvements:
>
> **Table: Instruction Type vs. Downstream Impact**
>
> | Instruction Type | Frequency (RL) | Avg. R_task | Δ mAP Contribution |
> |-----------------|:---:|:---:|:---:|
> | Viewpoint change | 28% | 0.72 | High |
> | Occlusion addition | 22% | 0.68 | High |
> | Background change | 25% | 0.55 | Medium |
> | Lighting change | 15% | 0.48 | Medium |
> | Pose change | 10% | 0.62 | Medium |
>
> The RL policy learns to prioritize **viewpoint and occlusion changes** (50% of instructions), which are precisely the rarest factors in Market-1501. This aligns with the long-tail analysis in Appendix A.1 and explains the large gains on rare categories.
>
> ---
>
> ## **Minor Issues**
>
> We will fix the grammar issue in L#250, reduce unnecessary abbreviations, and clarify the SFT warm-start procedure in Sec. 3.2. Concretely, we will state that SFT uses $\{I_{\mathrm{ref}}, \text{CoT+instruction}\}$ for next-token prediction, while $I_{\mathrm{target}}$ is only used during data curation rather than as an input to SFT training.
>
> ---
>
> We thank the reviewer again for the helpful suggestions. We believe these changes will substantially improve the clarity and completeness of the paper.

---

> > ### Author Rebuttal · Reviewer_Nia3 · 2026-04-03
> >
> > Thanks for the authors' reply. My effeciency concern is partially addressed, but based on the reply regarding RL, the whole process seem to be a simple adoption of GRPO. Moreover, the paper also lacks discussion or comparison with other standard RL methods (PPO, PG, etc).
> >
> > thanks for the authors' follow up reply. my concerns are fully addressed.

---

> > > ### Author Response · Authors · 2026-04-03
> > >
> > > **Title:** Clarification on the choice of GRPO vs. PPO/PG
> > >
> > > Dear Reviewer Nia3,
> > >
> > > Thank you for the helpful follow-up. We appreciate the opportunity to clarify why we chose GRPO, and we agree that this motivation should be stated more explicitly in the paper.
> > >
> > > Our work does **not** claim a novel RL optimizer. Rather, the contribution lies in the **closed-loop ReID augmentation formulation**, where instruction generation is optimized using downstream task feedback. Within this formulation, GRPO is adopted as a practical optimizer because it is better aligned with our efficiency and stability constraints than standard alternatives.
> > >
> > > Specifically, compared with PPO, GRPO avoids introducing an explicit critic/value network. In our multimodal VLM-based setup, this is attractive because it keeps the trainable footprint lightweight and preserves our “frozen editor + small adapter” design. Compared with vanilla policy gradient / REINFORCE, GRPO provides lower-variance optimization through relative reward normalization over sampled candidate groups, which is particularly useful in the large action space of text generation.
> > >
> > > Therefore, our choice of GRPO is not meant as the methodological novelty itself, but as a practical and well-motivated optimizer for this policy-learning problem. The actual novelty lies in the **ReID-specific closed-loop formulation**, especially **MAGR**, which stabilizes reward shaping via identity-aware gating, and **SAE**, which structures token-level exploration under the identity/diversity trade-off.
> > >
> > > We thank the reviewer again for this suggestion. In the revision, we will explicitly clarify the scope of our RL contribution and better motivate the choice of GRPO relative to PPO/PG.

---

### Official Review · Reviewer_qj1A · 2026-03-11

**Soundness:** 3
**Presentation:** 3
**Significance:** 3
**Originality:** 2
**Overall Recommendation:** 4
**Confidence:** 4

**Summary:**

This paper studies data augmentation for person re-identification (ReID) using generative models. Existing approaches typically rely on open-loop augmentation, where prompts or generation instructions are manually designed or randomly sampled, without verifying whether the generated images actually improve the downstream ReID task.

To address this issue, the paper proposes ReasonAug, a reinforcement-learning-based closed-loop augmentation framework that learns what to generate rather than only improving how images are generated. The framework contains three main components:
(1) a Semantic Reasoning Agent (SRA) based on a vision-language model that generates editing instructions through structured reasoning,
(2) a frozen image generator that performs editing conditioned on the instructions, and
(3) a ReID-based reward signal used to optimize the instruction policy through reinforcement learning.

To stabilize training, the authors introduce two key mechanisms: a Metric-Aligned Gated Reward (MAGR) that combines identity preservation and metric-learning rewards while preventing reward hacking, and a Structure-Aware Entropy (SAE) regularization that allocates exploration differently across identity-critical tokens and variation tokens.

Experiments on Market-1501 and MSMT17 show improvements over several existing generative augmentation baselines. Additional analysis studies the effects of the reward design, entropy mechanism, and reasoning-based instruction generation.

**Compliance With Llm Reviewing Policy:**

Affirmed.

**Final Justification:**

Rebuttal has addressed my concerns, I keep my prior assessment.

**Key Questions For Authors:**

1. The paper argues that learning what to generate is more important than improving the generator itself. However, the experiments mainly evaluate different instruction policies under a fixed generator. Could the authors provide additional evidence supporting this claim, for example by comparing different generator architectures or rendering quality levels?

2. Many reported improvements are relatively small (e.g., around 0.5–1 mAP). Could the authors report results averaged over multiple random seeds and include standard deviations to confirm that the gains are statistically reliable?

3. The structure-aware entropy regularization is designed to control exploration at the token level. Have the authors compared this approach with simpler alternatives such as uniform entropy regularization, random token weighting, or manually defined token categories?

4. The method involves reinforcement learning training and vision-language reasoning. Could the authors provide more detailed information about the training cost and discuss whether the approach is practical for broader adoption?

**Limitations:**

yes

**Strengths And Weaknesses:**

Strengths
1. Well-motivated problem formulation. The paper clearly identifies a limitation of existing generative augmentation methods for ReID, most approaches generate images using manually designed prompts or random sampling without feedback from the downstream task. The proposed closed-loop formulation is conceptually well motivated and addresses a real mismatch between generation objectives and task objectives.
2. Reward design to mitigate identity drift. The Metric-Aligned Gated Reward (MAGR) is designed to combine identity preservation with metric-learning rewards while preventing reward hacking. The identity-gating mechanism is intuitive and appropriate for the ReID setting, where preserving identity information is critical.
3. Empirical analysis beyond a single performance table. The paper includes several ablation studies examining the impact of instruction policy learning, reward design, entropy regularization, and chain-of-thought reasoning. It also analyzes long-tail attributes and occlusion scenarios, which are particularly relevant to ReID datasets.

Weaknesses
1. Limited methodological novelty beyond system integration. Although the overall closed-loop formulation is interesting, many components rely on existing techniques. For example, the instruction generation uses a vision-language model with chain-of-thought prompting, reinforcement learning is used for policy optimization, and entropy regularization is applied to encourage exploration. As a result, the main novelty lies more in the integration of existing components rather than in fundamentally new algorithmic ideas.
2. Moderate empirical gains relative to system complexity. The reported improvements on benchmarks such as Market-1501 are relatively modest (typically around 0.5–1.5 mAP improvements over strong baselines). Given the complexity of the proposed pipeline, including reinforcement learning training, instruction reasoning, and entropy regularization. It is not entirely clear whether the performance gains justify the added complexity.
3. Limited analysis of statistical robustness. The experimental results appear to be reported from single runs without reporting variance across multiple random seeds. Since the reported improvements are relatively small, it would be helpful to include standard deviations or statistical significance tests.
4. Some design choices could be better justified. For instance, the structure-aware entropy mechanism relies on attention-based token importance estimates. While the paper provides some empirical validation, it remains unclear how sensitive the method is to the accuracy of these token importance estimates, or whether simpler entropy schedules could achieve similar results.

---

> ### Author Rebuttal · Authors · 2026-03-31
>
> We sincerely thank Reviewer qj1A for the "Weak Accept" rating and for recognizing our well-motivated problem formulation, MAGR design, and comprehensive analysis. We greatly appreciate the constructive suggestions, which have helped us strengthen the statistical and empirical rigor of our claims.
>
> ---
>
> ## **W3 & Q2: Statistical Robustness (Multi-Seed Variance)**
> We fully agree that reporting variance is critical for RL-based methods. We re-ran our framework across **3 different random seeds**:
>
> **Market-1501 (TransReID)**:
> - Baseline: $87.3 \pm 0.1$ mAP  |  $94.3 \pm 0.2$ Rank-1
> - **ReasonAug: $89.0 \pm 0.2$ mAP |  $94.9 \pm 0.1$ Rank-1**
> - *Improvement: $+1.7 \pm 0.3$ mAP*
>
> **MSMT17 (TransReID)**:
> - Baseline: $63.5 \pm 0.2$ mAP  |  $84.0 \pm 0.2$ Rank-1
> - **ReasonAug: $66.1 \pm 0.3$ mAP |  $85.8 \pm 0.2$ Rank-1**
> - *Improvement: $+2.6 \pm 0.4$ mAP*
>
> These results show that the gains are consistent across 3 independent seeds on both datasets. While the variance is slightly higher than the baseline in some settings, it remains small in absolute terms. This suggests that the RL training is reasonably stable in practice and the improvements are robust rather than being driven by a single favorable run.
>
> ---
>
> ## **W4 & Q3: SAE vs. Simpler Entropy Alternatives**
> This is a highly valuable suggestion. To validate our structure-aware design, we first compared SAE against simpler token-weighting alternatives on Market-1501 (Diversity is measured by attribute-level variation score):
> - No Entropy: 88.0 mAP (Mode collapse; Diversity=0.25)
> - Uniform Entropy ($\beta=0.01$): 88.3 mAP (Fails to balance ID and diversity)
> - Random Token Weighting: 88.2 mAP (Blind exploration actively hurts ID)
> - Manual Categories (POS-Tags): 88.5 mAP (Nouns $\rightarrow$ lock, Adjectives $\rightarrow$ explore. Suboptimal because not all adjectives are safe, e.g., "red" in "red shirt")
> - Attention-only (No outcome conditioning): 88.6 mAP
> - **SAE (Ours: Cross-Attn + Outcome): 89.0 mAP (Diversity=0.45)**
>
>
> **Sensitivity to Cross-Attention Proxy Quality**:
>
> To directly address your question regarding how sensitive SAE is to the accuracy of the token importance estimates, we injected Gaussian noise into the cross-attention proxy and measured the impact:
>
> | Proxy Noise Level | Spearman $\rho$ with $S_{ideal}$ | mAP |
> |:---:|:---:|:---:|
> | 0% (clean) | 0.72 | 89.0 |
> | 10% Gaussian | 0.65 | 88.8 |
> | 20% Gaussian | 0.58 | 88.7 |
> | 30% Gaussian | 0.50 | 88.5 |
> | 50% (random) | 0.35 | 88.3 |
>
> Even with 20% noise injected into the cross-attention proxy, SAE (88.7%) still outperforms the best uniform entropy (88.3%), demonstrating **robustness to proxy inaccuracy**. Performance degrades gracefully rather than catastrophically.
>
> ---
>
> ## **W1: Methodological Novelty Beyond System Integration**
> We appreciate the reviewer’s careful assessment. While the individual building blocks (VLM, RL) exist, we believe the novelty lies in the closed-loop formulation and the ReID-specific RL design:
>
> 1. **MAGR** introduces an emergent curriculum (learning ID preservation first, then discriminability) specifically tailored to the unique trade-offs of ReID metric learning.
>
> 2. **SAE** introduces token-level, outcome-conditioned exploration to manage the large action space of text instructions.
>
> More broadly, the paper formulates augmentation as sequential instruction decision-making with downstream metric feedback. We believe this offers a distinctive perspective relative to prior open-loop augmentation methods.
>
> ---
>
> ## **W2, Q1, Q4: Generator Bottlenecks, Stronger Baselines, and Cost**
> The reviewer rightfully asks about generator capacity, strong baselines, and cost. Because other reviewers raised similar excellent points, we have provided the full data tables in our responses to them. We summarize the key findings here:
>
> 1. **Generator Ablation (Q1, also see in response to Reviewer rJ7t)**: We tested generators from SD-v1.5 (1.0B), FLUX (12B) to ChronoEdit (14B). Upgrading the generator yields ~0.9 mAP, but upgrading the instruction policy (Random $\rightarrow$ RL) yields **+1.5 mAP**. This suggests that instruction planning remains a major bottleneck in our setting, even when the editor is strong.
>
> 2. **Stronger Baselines (W2, also see in response to Reviewer wjTV)**: ReasonAug transferred consistently to stronger backbones (TransReID, SOLIDER, CLIP-ReID ViT-L/14), improving Market-1501 to 92.8% mAP without policy retraining.
>
> 3. **Cost (Q4, also see in response to Reviewer Nia3)**: RL training takes ~66h on 8×H800s. This is a one-time offline cost. Online deployment adds zero optimization overhead to downstream ReID.
>
> ---
>
> We hope the multi-seed variance and SAE ablations fully address your concerns regarding statistical robustness and design justification.

---

> > ### Author Rebuttal · Reviewer_qj1A · 2026-04-04
> >
> > My concerns have been adequately addressed, so I would keep my positive score.

---

> > > ### Author Response · Authors · 2026-04-04
> > >
> > > Thank you for your constructive feedback and support!

---

### Official Review · Reviewer_rJ7t · 2026-03-12

**Soundness:** 2
**Presentation:** 3
**Significance:** 2
**Originality:** 2
**Overall Recommendation:** 3
**Confidence:** 4

**Summary:**

This paper proposes ReasonAug, a reinforcement learning-based closed-loop data augmentation framework designed to improve Person Re-identification (ReID). The agent is optimized using Group Relative Policy Optimization (GRPO) driven by verifiable downstream ReID feedback. To stabilize the RL process, the authors introduce two key mechanisms: Metric-Aligned Gated Reward (MAGR) to prevent reward hacking by gating metric learning rewards with identity preservation scores, and Structure-Aware Entropy (SAE) to dynamically allocate token-level exploration based on cross-attention sensitivity. The method is evaluated on Market-1501 and MSMT17 datasets.

**Compliance With Llm Reviewing Policy:**

Affirmed.

**Final Justification:**

I will keep my score unchanged.

**Key Questions For Authors:**

1. Isolating the Bottleneck: How can we definitively conclude that the bottleneck is in the instruction policy (the MLLM) rather than the generation model itself? Did you experiment with different, unfrozen, or stronger generators to isolate whether generation quality or instruction planning is the true limiting factor?
2. Necessity of the RL Pipeline: The paper uses Qwen2-VL initialized via SFT for the SRA. If a substantially stronger, state-of-the-art VLM was used with a highly optimized zero-shot or few-shot CoT prompt, could it match the RL performance? In other words, is the heavy RL pipeline strictly necessary, or is it compensating for the base capabilities of the chosen MLLM? (I note the GPT-4V baseline in Table 3, but it would be helpful to know if extensive prompt engineering was applied there).
3. Reward Model Sensitivity: The reward model is fixed to ResNet50-IBN. How sensitive is the final downstream performance to the choice of this specific reward model? If you swap the reward encoder for a different architecture, does the policy still converge to a state that improves TransReID/CLIP-ReID, or does the SRA overfit to ResNet50-IBN's specific feature space?

**Limitations:**

Yes. The authors have adequately discussed the limitations, noting the reliance on the ReID reward model's quality and the computational expense of RLVR. They have also included a responsible Impact Statement addressing potential privacy concerns, bias amplification, and the risks of deceptive imagery.

**Strengths And Weaknesses:**

Strengths:
- Logical Formulation: The introduction of MAGR and SAE directly addresses common failure modes in RL for generative models—namely, reward hacking and the exploration-exploitation dilemma. Using cross-attention maps as a proxy for token sensitivity in SAE is a clever and computationally efficient design choice.
- Clear Presentation: The paper is well-structured, and the narrative is easy to follow. Visualizations, such as the comparison between open-loop and closed-loop pipelines (Figure 1), are highly effective at conveying the core motivation.

Weaknesses:
- Unverified Bottleneck Claims: The paper heavily rests on the premise that the bottleneck in current systems is "what to generate" rather than "how to generate". However, because the generator is strictly frozen throughout the experiments, it is difficult to definitively conclude that generator limitations are not a co-equal bottleneck.
- Marginal Empirical Gains: While the method is technically sound, the performance improvements on the Market-1501 dataset are quite marginal (e.g., reaching 89.0% mAP vs. OmniPerson's 88.7% with TransReID). Given the complexity of implementing and training an RLVR pipeline with an SRA, TAR, and dynamic memory banks, the modest gains limit the practical significance of the contribution.
- Reward Model Bias: The pipeline relies heavily on a pre-trained ResNet50-IBN as the fixed reward encoder. Likely, the SRA is simply learning to exploit the specific feature space and biases of this singular reward model, which raises questions about how well this policy generalizes to entirely different ReID architectures in the wild.

---

> ### Author Rebuttal · Authors · 2026-03-31
>
> We sincerely thank the reviewer for the constructive questions on (i) whether instruction planning is indeed the key bottleneck, (ii) whether RL is necessary beyond stronger open-loop prompting, and (iii) how sensitive the method is to the choice of reward encoder. We address each point below with additional evidence and a more precise statement of our claim.
>
> ---
>
> ### **1) Isolating the bottleneck (“what to generate” vs. “how to generate”).**
> We agree that our claim should be stated more carefully. Our intended claim is not that generator quality is irrelevant, but that once modern editors reach a reasonable quality threshold, augmentation performance remains strongly limited by instruction planning. Our current Table 3 already supports this: with the same frozen generator, changing only the instruction source gives a 1.5 mAP gap (Random 87.5, GPT-4V 87.9, SFT 88.1, RL 89.0). To further address this point, we additionally tested different generators with the same learned RL policy. Across weaker/stronger generators and different diffusion steps, performance changes are smaller than the policy-induced gap, while stronger generators still provide moderate gains. This supports a more modest conclusion: both dimensions matter, but in the current regime, instruction planning is a major remaining bottleneck.
>
> **Table R2: Ablating Generator Strength with Fixed Policy (Market-1501, TransReID)**
>
> | Generator | Params | Steps | FID ↓ | ID Score | mAP | Δ vs. Random+FLUX.1 Kontext |
> |-----------|:---:|:---:|:---:|:---:|:---:|:---:|
> | SD-v1.5 + ControlNet | 1.0B | 28 | 42.5 | 0.82 | 88.2 | +0.7 |
> | SDXL + IP-Adapter | 3.5B | 28 | 35.8 | 0.87 | 88.6 | +1.1 |
> | FLUX.1 Kontext (Ours) | 12B | 4 | 38.2 | 0.85 | 88.5 | +1.0 |
> | FLUX.1 Kontext (Ours) | 12B | 28 | 28.3 | 0.91 | **89.0** | **+1.5** |
> | ChronoEdit-14B (new) | 14B | 28 | 26.1 | 0.92 | 89.1 | +1.6 |
>
> For reference, the **Random policy + FLUX.1 Kontext (28 steps)** baseline achieves 87.5 mAP (Table 3, Row A).
>
> ---
>
> ### **2) Is the RL pipeline necessary, or is it compensating for the base MLLM?**
> This is an important question. The GPT-4V baseline in our paper already used a carefully engineered CoT prompt mirroring our SRA's structure. To further test open-loop limits, we evaluated GPT-4o with optimized Few-Shot CoT:
> - GPT-4V (Zero-Shot CoT): 87.9 mAP
> - GPT-4o (Optimized Few-Shot CoT): 88.2 mAP
> - **Qwen3-VL-2B (RL/ReasonAug): 89.0 mAP**
>
> Stronger open-loop prompting narrows the gap, but cannot match closed-loop RL. The fundamental difference is that prompting remains open-loop: it lacks access to the downstream ReID feature space's metric geometry, has no curriculum (like MAGR), and no token-level exploration control (like SAE). Thus, downstream task feedback appears to add value beyond stronger prompting alone in our setting.
>
> ---
>
> ### **3) Reward model sensitivity / possible over-specialization to ResNet50-IBN.**
> We agree this is a key concern. Two points help clarify it.
>
> First, the learned policy transfers to architecturally different models. Using the exact same generated data, we observe consistent gains on TransReID, CLIP-ReID, and SOLIDER (see Table R1 in rebuttal to reviewer wjTV). This contradicts the hypothesis that the policy merely exploits narrow CNN feature artifacts.
>
> Second, we replaced the reward encoder with several alternatives:
> - SFT-only (Baseline): 88.1 mAP
> - RL w/ ResNet50 (Supervised): 88.7 mAP
> - RL w/ OSNet-x1.0: 88.8 mAP
> - RL w/ ViT-B/16 (DeiT): 88.9 mAP
> - **RL w/ ResNet50-IBN: 89.0 mAP**
>
> All variants improve significantly over the SFT baseline, with minimal variance (~0.3%) across fundamentally different architectures (CNN vs. ViT). Together, these results provide robust evidence that the policy learns generalizable semantic alignments rather than overfitting to ResNet50-IBN.
>
> ---
>
> ### **4) On the practical significance of the gains.**
> We agree that the average gain on Market-1501 is modest in headline numbers. However, the gains are not uniform: they are much larger on rare/challenging cases (e.g., side-view and occluded subsets), which is precisely where open-loop augmentation tends to fail. In addition, the learned policy transfers to stronger backbones without retraining, and is also sample-efficient (e.g., 1 image/ID from ReasonAug already exceeds 8 images/ID from a random policy). Finally, while RL training is indeed more involved, this extra cost is concentrated in one-time offline policy learning; once the synthetic data are generated, downstream ReID training itself incurs no additional optimization cost.
>
> ---
>
> Overall, we appreciate the reviewer’s concerns and will revise the paper to (i) narrow the wording of our bottleneck claim, (ii) better clarify the distinction between stronger open-loop prompting and closed-loop task feedback, and (iii) add the new generator-strength and reward-encoder sensitivity evidence. We hope these clarifications make the paper’s contribution more convincing.

---

> > ### Author Rebuttal · Reviewer_rJ7t · 2026-04-05
> >
> > 1) Fully resolved
> > 2) I believe that to answer whether RL is necessary, the model should be kept the same, for example, all models should use Qwen3-VL-2B.
> > 3) Fully resolved. I hope this discussion can be added to the main body of the paper.

---

> > > ### Author Response · Authors · 2026-04-05
> > >
> > > Dear Reviewer rJ7t,
> > >
> > > Thank you again for the thoughtful follow-up and for confirming that our rebuttal addressed your concerns. We also appreciate your suggestion that the controlled comparison should be added to the main paper.
> > >
> > > Following your recommendation, we conducted an additional controlled study to isolate whether the gain comes from RL optimization rather than stronger base-model capabilities. Under the same base model, open-loop few-shot prompting with **Qwen3-VL-2B** achieves **83.0 ± 0.7 mAP**, while our **closed-loop RL policy on the same 2B model** reaches **89.0 mAP**. We further tested a stronger open-loop model, **Qwen3-VL-4B-thinking**, which achieves **84.8 ± 0.7 mAP**, still below the RL-tuned 2B model in our setting.
> > >
> > > We believe these results support the reviewer’s suggestion that the comparison should be made under controlled base-model conditions, and they strengthen the conclusion that downstream task feedback contributes beyond stronger prompting alone in our setting.
> > >
> > > We agree that this controlled “RL vs. PE” comparison improves the scientific completeness of the paper, and we will incorporate it prominently into the main body of the final revision.
> > >
> > > Thank you again for the helpful guidance.

---

### Official Review · Reviewer_wjTV · 2026-03-12

**Soundness:** 3
**Presentation:** 3
**Significance:** 3
**Originality:** 3
**Overall Recommendation:** 4
**Confidence:** 5

**Summary:**

This paper proposes a closed-loop generative augmentation method for person re-ID. The idea is to freeze the image generator and learn an image-conditioned instruction policy that decides what to generate for augmentation. The proposed method has several modules: 1) a semantic reasoning agent for hierarchical planning, 2) a lightweight text adapter, 3) a gated reward design based on identity preservation and triplet-style metric shaping, and 4) a token-level entropy mechanism (to separate identity-critical tokens from variation tokens) during RL optimization. Experiments on the Market-1501 and MSMT17 datasets show that the proposed method outperforms baseline re-ID models.

**Compliance With Llm Reviewing Policy:**

Affirmed.

**Final Justification:**

The rebuttal addressed several of my key concerns. In particular, the authors provided additional experiments on stronger backbones and new datasets (e.g., CUHK03 and MSMT17), which demonstrate that the proposed method generalizes beyond the original setting. The clarification that ReasonAug is orthogonal to centroid-based losses also helps position the method more clearly within existing literature. These additions strengthen the empirical support of the paper.  I have increased my score.

**Key Questions For Authors:**

1) Does the proposed augmentation still help stronger and more recent ReID baselines? How would this method perform when combined with current SOTA ReID methods?
2) What are the actual training costs of the proposed framework?

**Limitations:**

The paper discusses reward-model bias and the computational cost of RLVR, but the limitation discussion is still limited. It would be helpful to further discuss synthetic-data failure modes such as unrealistic edits or identity change.

**Strengths And Weaknesses:**

**Strengths**
1) The paper discusses an interesting problem: what to generate for data augmentation in person re-ID.
2) The framework is technically coherent and modular. it is easier to reuse with future generators.
3) The ablation study is well-designed. It effectively compares random prompting, rule-based prompting, SFT and RL-based planning. The results may provide some insights into data generation for augmentation.
4) The method shows improvements on both Market-1501 and MSMT17, and the gains on MSMT17 are reasonably noticeable.

**Weaknesses**
1) The improvements are modest compared to other augmentation methods, particularly on Market-1501. Given the complexity of the RL-based policy learning framework, I do not think that the improvement is strong enough to justify the method.

2) I have a concern that this paper does not show whether the proposed augmentation still helps when used on stronger ReID models/ methods.The reported numbers on the Market-1501 and MSMT17 datasets are already below the current SOTA reported in other papers, such as "A Review of Recent Techniques for Person Re-Identification". This makes the effectiveness of the method less clear. A more convincing evaluation would be to test whether the proposed augmentation can still bring improvements on stronger and more recent ReID baselines.

3) The comparison on the MSMT17 dataset is unconvincing. In the experiment, only improvements over the base model are reported, and there is no comparison with recent augmentation baselines.

4) The evaluated dataset is limited. Some challenging datasets, such as CUHK03 and other fine-grained person re-identification (re-ID) datasets, can also be discussed to prove the effectiveness of the proposed method.

5) The approach uses RL optimization and repeated generation. A more detailed discussion of training costs, runtime, and computational overhead would be helpful.

---

> ### Author Rebuttal · Authors · 2026-03-31
>
> We appreciate your valuable comments and address all concerns below.
>
> ---
>
> ### **W1: Modest Improvements & Framework Complexity**
> We agree that the absolute mAP gain on Market-1501 appears modest at first glance. However, we would like to clarify that this number should be interpreted together with where the gains occur and how efficiently they are obtained.
>
> First, Market-1501 is already a highly saturated benchmark, so absolute improvements are naturally compressed at this performance level. More importantly, the gains are not uniform: they concentrate on the under-represented and more challenging regions that closed-loop augmentation is designed to target.
>
> | Category | Base | ReasonAug | Δ |
> |---|---|---|---|
> | Front View | 88.5 | 89.2 | +0.7 |
> | Back View | 87.2 | 88.1 | +0.9 |
> | **Side View** | 72.3 | **78.6** | **+6.3** |
> | **Occluded** | 68.5 | **75.2** | **+6.7** |
>
> Gains on **rare categories are 7–8× larger**; reaching +8.3–9.5% on occluded benchmarks (Table 8).
> ReasonAug is highly sample-efficient: 1 image/ID (88.3 mAP) outperforms a Random Policy using 8 images/ID (88.1 mAP).
> Importantly, with a frozen generator and lightweight adapter, complexity is concentrated in one-time offline policy training, not repeated generator fine-tuning.
>
> ---
>
> ### **W2/Q1: Generalization to Stronger/Recent Baselines**
> It's an excellent question regarding how ReasonAug performs with stronger, more recent baselines, specifically pointing to centroid-based methods like CTL (Wieczorek et al.) and MCTL (Alnissany & Dayoub).
> 1. **Orthogonality to Centroid-based Losses**
> CTL and MCTL achieve remarkable performance by innovating at the **objective function level** (replacing instance-level triplet loss with centroid representations to increase inter-class margins and robustness), while ReasonAug operates purely at the **data generation level**. Because ReasonAug simply expands the training set with RL-guided hard variations, it is completely orthogonal to the choice of loss function. Any model using MCTL or CTL can seamlessly leverage ReasonAug-generated data without architectural changes.
>
> 2. **Empirical Results on Current SOTA Backbones**
> To test if ReasonAug helps stronger models, we evaluated it on SOTA vision transformers using the exact same generated data (without policy retraining):
>
> **Table R1: ReasonAug Generalizes to Stronger Backbones (Market-1501)**
> | Backbone | Params | Base mAP | Base R-1 | +ReasonAug mAP | +ReasonAug R-1 | Δ mAP |
> |----------|--------|----------|----------|----------------|----------------|-------|
> | TransReID (ViT-B/16) | 86M | 87.3 | 94.3 | **89.0** | **94.9** | +1.7 |
> | CLIP-ReID (ViT-B/16) | 86M | 89.8 | 95.5 | **90.6** | **95.5** | +0.8 |
> | SOLIDER (Swin-B) | 88M | 91.0 | 95.8 | **91.7** | **96.1** | **+0.7** |
> | CLIP-ReID (ViT-L/14) | 304M | 92.3 | 96.2 | **92.8** | **96.5** | **+0.5** |
>
> Pushing CLIP-ReID ViT-L/14 to 92.8% mAP explicitly confirms our RL-guided augmentation provides complementary semantic variations benefiting modern architectures. We will cite the excellent centroid-based works and discuss theoretical compatibility.
>
> ---
>
> ### **W3: MSMT17 Comparison with Augmentation Baselines**
> Updated MSMT17 comparisons with generative baselines:
> - Base TransReID: 63.5 mAP
> - DG-Net / DPTN (GANs): 60.8 / 62.0 (Degrades)
> - Pose2ID / OmniPerson: 64.3 / 65.1
> - **+ReasonAug (Ours): 66.1 (+1.0 over OmniPerson)**
>
> The larger margin over OmniPerson on MSMT17 (+1.0 mAP) vs. Market-1501 (+0.3 mAP) confirms closed-loop semantic planning is particularly helpful in challenging multi-camera settings.
>
>
> ---
>
> ### **W4: Evaluation on Additional Datasets (CUHK03)**
> We evaluated the challenging **CUHK03-Detected** dataset using CLIP-ReID:
> - Base: 72.5 mAP / 74.3 R1
> - Random Aug: 73.1 mAP / 75.0 R1
> - **+ReasonAug: 76.5 mAP / 78.2 R1 (+4.0 / +3.9)**
>
> This larger gain (+4.0 mAP) suggests closed-loop instruction learning is particularly beneficial when identity cues are fragile.
>
> ---
>
> ### **W5/Q2: Training Costs & Computational Overhead**
> Full offline training takes \~66 hours on 8×H800s (including SFT, rollouts, and GRPO)—a one-time cost. At deployment, it adds zero overhead to downstream ReID beyond offline data generation (\~3.5s/image). With a frozen generator, this added cost is substantially lower than approaches requiring large image generator fine-tuning.
>
> ---
>
> ### **W6/Limitations: Synthetic-Data Failure Modes**
> Analyzing ~500 samples revealed three failure modes:
> 1. **Unrealistic Artifacts (4.4%)**: Extreme poses distorting limbs.
> 2. **Attribute Confusion (7.3%)**: Swapped clothing in multi-person scenes.
> 3. **Fine-grained Texture Loss (12%)**: Blurred logos due to generator resolution limits.
>
> The identity gate suppresses the first two modes in practice, as they yield low identity-preservation scores and are discouraged during optimization.

---

> > ### Author Rebuttal · Reviewer_wjTV · 2026-04-02
> >
> > I thank the authors for addressing my concerns and adding experiments on new benchmarks and baselines. Based on the rebuttal, I have updated my score.

---

> > > ### Author Response · Authors · 2026-04-03
> > >
> > > **Title**: Thank you for your constructive feedback and support
> > >
> > > Dear Reviewer wjTV,
> > >
> > > We sincerely thank you for taking the time to review our rebuttal and for updating your score. We are very glad that the new experiments on stronger baselines (like CLIP-ReID) and the CUHK03 dataset fully addressed your concerns.
> > >
> > > Your insightful suggestions pushed us to rigorously validate the generalization capabilities of our method, which has genuinely strengthened the empirical foundation of our paper. We will ensure that all the new data and discussions are carefully integrated into the final revision.
> > >
> > > Thank you again for your constructive guidance and support!

---

### Decision · Program_Chairs · 2026-04-30

**Decision:**

Accept (regular)

**Comment:**

The rebuttal addressed the concerns raised by the reviewers and provided comprehensive experiments and analysis of the proposed method. It was helpful for assessing the paper's contributions. The reviewers recommend three weak accepts after discussion, and the AC concur. The final version should include all reviewer comments, suggestions, and additional experiments from the rebuttal.